

**Effects of dynamic changes of desiccation cracks on preferential flow: Experimental investigation and**
**numerical modeling**
Yi Luo[1,2], Jiaming Zhang*[1], Zhi Zhou[3], Juan P. Aguilar-Lopez[4], Roberto Greco[5], Thom Bogaard[2]
[1] Faculty of Engineering, China University of Geosciences (Wuhan), Wuhan, 430074, China
[2] Water Resources Section, Faculty of Civil Engineering and Geosciences, Delft University of Technology, Stevinweg 1, P.O.
Box 5048, 2600 GA Delft, the Netherlands
[3.]Department of Engineering Management, Hubei University of Economics, Wuhan, 430205, China
[4.]Department of Hydraulic Engineering, Faculty of Civil Engineering and Geosciences, Delft University of Technology, Delft,
2600 GA, the Netherlands
[5.]Dipartimento di Ingegneria, Università degli Studi della Campania "Luigi Vanvitelli", via Roma 29, 81031, Aversa, Italy
*Correspondence to*: Jiaming Zhang (zjm@cug.edu.cn)
**Abstract:** Preferential flow induced by desiccation cracks (PF-DC) has been proven to be an important hydrological effect
that could cause various geotechnical engineering and ecological environment problems. Investigation on the PF-DC remains
a great challenge due to the soil shrinking-swelling behavior. This work presents an experimental and numerical study of the
PF-DC considering the dynamic changes of DC. A soil column test was conducted under wetting-drying cycles to investigate
the dynamic changes of DC and their hydrological response. The ratio between the crack area and soil matrix area (crack
ratio), crack aperture and depth were measured. The soil water content, matrix suction and water drainage were monitored. A
new dynamic dual-permeability preferential flow model (DPMDy) was developed, which includes physically-consistent
functions in describing the variation of both porosity and hydraulic conductivity in crack and matrix domains. Its performance
was compared to the single-domain model (SDM) and rigid dual-permeability model (DPM) with fixed crack ratio and
hydraulic conductivity. The experimental results showed that the maximum crack ratio and aperture decreased when the
evaporation intensity was excessively raised. The self-closure phenomenon of cracks and increased surficial water content
were observed during low evaporation periods. The simulation results showed that the matrix evaporation modeled by the
DPMDy is lower than that of the SDM and DPM, but its crack evaporation is the highest. Compared to the DPM, the DPMDy
simulated a faster pressure head building-up process in the crack domain and higher water exchange rates from the crack to
the matrix domain during rainfall. Using a fixed crack ratio in the DPM, whether it is the maximum or the average value from
the experiment data, will overestimate the infiltration fluxes of PF-DC but underestimate its contribution to the matrix domain.
In conclusion, the DPMDy better described the underlying physics involving crack evolution and hydrological response with
respect to the SDM and DPM. Further improvement of the DPMDy should focus on the hysteresis effect of the SWRC curve
and soil deformation during wetting-drying cycles.
**Keywords:** Desiccation cracks; preferential flow; dynamic changes; dual-permeability model; wetting-drying cycles
**1.   Introduction**
Desiccation cracks are prevalent in clay-dominated soils due to water loss, which often lead water to bypass the surface soil
matrix and rapidly infiltrate into subsoil as preferential flow (Davidson, 1984; Weiler, 2005). Positively, the preferential flow
induced by desiccation cracks (PF-DC) can promote the migration of farmland organic matter (Vervoort et al., 2003) and
reduce surface runoff (Pei et al., 2020; Zhang et al., 2021a). Negatively, it also has proven to be an important hydrological



mechanism that could lead to geotechnical engineering and ecological environment problems, such as dike and slope
instability (Jamalinia et al., 2020; Zhang et al., 2021b), shallow landslides (Bogaard and Greco, 2015; Caris and Van Asch,
1991; Luo et al., 2021), groundwater pollution (Chaduvula et al., 2022; Chen et al., 2002; Mooney and Morris, 2008; Schlögl
et al., 2022) and reduction of irrigation efficiency (Greve et al., 2010; Smith et al., 2005; Wang et al., 2018; Wang et al., 2022).
Under the current background of frequent extreme flood-drought climate events, its negative effects will be more prominent
(Tichavsky et al., 2019). Investigation on the PF-DC are of great significance in guiding scientific research and practical
design in the above disciplines.
A unique characteristic of the desiccation cracks is their dynamic features, often causing instantaneous variation of crack
proportion, depth and connectivity with moisture content. Previous efforts have attempted to reveal the effects of crack
dynamics on the PF-DC through experiment studies, but most of them focused on short-term wetting process and obtained
only qualitative results and debates remained. For instance, Favre et al. (1997)and Liu et al. (2003) stated that crack closure
due to wetting can cause a significant reduction or even disappearances in the preferential flow. However, other studies found
that the PF-DC also leads water to rapidly infiltrate into deep soil even desiccation cracks are nearly closed (Baram et al.,
2012a; Greve et al., 2010; Luo et al., 2021; Tuong et al., 1996; Sander and Gerke, 2007). Cheng et al. (2021) conducted a
series of constant-head permeability tests with the hydraulic head gradient of 15 kPa. They stated that 4% of surface crack
ratio could be a critical value for determining whether desiccation cracks cause a significant increase in the infiltration rate or
not. However, this value may vary with different soils, rainfall patterns and sample scales, and thus lacks general applicability.
Indeed, PF-DC has long-term and complex spatiotemporal variability due to crack dynamics during wetting-drying cycles.
Therefore, short-term and small-scale infiltration tests (i.e. laboratory permeability tests) are not enough to reveal the complex
hydrological process induced by PF-DC. Meanwhile, it is also difficult to quantitatively study PF-DC only through
experiments. An improve understanding of the PF-DC combined with theory methods is also needed.
Regarding the theoretical methods, explicit crack models (EMs) (Hendrickx. and Flury, 2001; Khan et al., 2017; Xie et al.,
2020)), dual-porosity (DPoM) (Van Genuchten, 1980; Van Genuchten and Wierenga, 1976) and dual-permeability (DPM)
(Aguilar‑López et al., 2020; Gerke and Van Genuchten, 1993b, 1993a) models were developed to simulate preferential flow
in cracked clay soils. EMs were constructed based on the single-domain (or single-permeability) framework, which require
to define the details involving the geometry, spatial distribution and hydrological properties of each crack. Such requirement
may be conceptually correct but makes them difficult for simulating network-distributed desiccation cracks due to
considerable computational burden (Aguilar‑López et al., 2020). The DPoM and DPM concepts belong to the dual-domain
framework that assumes the soil pore system can be represented as two overlapping interacting regions, one which represents
the matrix domain with micropores and the other one represents the crack domain with meso-macro pores (Šimůnek et al.,
2003). Those models represent the cracks in the soil as implicit form which need not to prescribe geometrical and spatial
features of the desiccation cracks. The DPoM concept holds the simplifying stipulation that water only flows through the
shrinkage cracks rather than the soil matrix, which is unrealistic in many cases. To remedy this shortcoming, classical DPM
was developed, where, the water flow in soil matrix and crack domain was simulated using the Richards' equation (Aguilar‑
López et al., 2020; Coppola et al., 2012; Gerke and Maximilian Köhne, 2004; Gerke and Van Genuchten, 1993a) or Green-
Ampt model (Davidson, 1984; Stewart, 2019; Weiler, 2005) building on Darcy's law. However, some critics emerged that
the Richards' equation building on the capillarity, not existing in large PF paths (e.g. tensile cracks and biological holes), is
not suitable to simulate the PF (Larsbo and Jarvis, 2003; Nimmo, 2010; 2021). Consequently, some improved DPMs were
developed, where, water flow in the crack domain was simulated by the Navier-Stokes equation (Germann and Karlen, 2016;
Nimmo, 2010), kinematic wave equation (Greco, 2002; Larsbo and Jarvis, 2003) and Poiseuille model (Lepore et al., 2009).
Although these improved DPM models better captured the characteristics of the water flow in the crack domain, the classical
DPM concept has still been widely accepted and used in simulating preferential flow in soils due to its easily available





parameters, reasonably satisfactory prediction to the measurements and high computation efficiency (Jarvis et al., 2016). Most
importantly, a recent numerical study conducted by Aguilar‑López et al. (2020) proved that effective parameter selection in
the DPM models can achieve similar modeling results to the EMs.
Nevertheless, classical DPM models often adopt the assumption that crack volume and hydrological properties keep constant
in both time and space, which is unfeasible to capture the full dynamics of PF-DC. Some attempts have been made to
incorporate the dynamic nature of desiccation cracks into DPM including the SWAP family of models, i.e. LEACHM, which
simulates PF-DC using a shrinkage characteristic and water loss (Kroes et al., 2000), but neglects water exchange process
occurring at the interface between two domains. Such a process has widely been confirmed to be significant in cracked soils
(Greve et al., 2010; Krisnanto et al., 2016; Tuong et al., 1996). Later modification of SWAP incorporated the aforementioned
process, but with a cost of neglecting shrink-swell behavior of soil. The VIMAC model developed by Greco (2002) solved
previous problems but against the cost of inducing many parameters which are difficult to determine from experiments or
measurements. Coppola et al. (2012); (2015) took another step forward to allowed crack volume and/or hydrological
properties to vary as a function of soil shrinkage. However, the relationship proposed in the model, a natural logarithm function
involving the suction head and crack proportion, lacks physical consistency with the variation of porosity. This implies a
disconnection between hydrological properties and porosity in the crack domain. Stewart et al. (2016b) deduced a shrinking-
swelling model, with relatively clear physical meaning and high consistency, and recently incorporated it into a Green-Ampt
based DPM (Stewart, 2018). While an analytical solution was obtained, the intrinsic limitation of the Green-Ampt approach
(i.e. hypothesis of the wetting front and request for a constant boundary condition) hindered the further application of this
model in complicated scenarios.
The objective of this research was to investigate the PF-DC from the experimental perspective in combination with an
effective modelling approach. Hence, a soil column test was conducted to investigate the dynamic changes of desiccation
cracks and hydrological response. The variation of crack geometry, including crack ratio, width and depth were measured.
The soil moisture content, matrix suction and water drainage were also monitored. Meanwhile, we developed a dynamic dual-
permeability preferential flow model by incorporating the shrinking-swelling model proposed by Stewart et al. (2016b). The
performance of the model was evaluated by comparing the simulated results with measured data.
**2.   Experimental study**
2.1 Testing apparatus
To investigate the effects of dynamic changes of desiccation cracks on preferential flow, a soil column infiltration test was
conducted under wetting-drying cycles (abbreviated as WD cycles hereafter). The testing apparatus consisted of a rainfall-
evaporation system, environment monitoring device, a plexiglass column, HD camera, hydrological sensors and drainage
measurement device (Fig. 1).
The rainfall-evaporation system included a rainfall simulator and two warm lamps as well as a small fan. The rainfall simulator
was 0.5 m above the soil surface, which can produce rainfall with the intensity of 24-120 mm/h. The warm lamps and a small
fan were put near the soil surface to accelerate water evaporation. The environment monitoring device consisted of a thermo-
hygrometer that connected a probe above the soil surface to detect the environmental temperature and humidity, and a water
container to measure the potential evaporation.
The plexiglass column was composed of a column (with a height of 60 cm and a diameter of 50 cm) placed on a catchment
hopper which was used to collect and drain out water from the soil column.
HD camera (TTQ-J2, constant focal length: 35 mm) was fixed on the slope above the soil surface to take photos at regular
intervals during the drying periods.
Hydrological sensors, including 5 soil moisture content/temperature sensors (Acclima, TDR-310s, with a measurement





moisture content range of 0-100%, an accuracy of ±2%; temperature range of -40 ℃ - +60 ℃, an accuracy of ±0.2 ℃) and 5
water potential sensors (Campbell, WP-257, with a measurement range of -200 kPa - 0 kPa, an accuracy of ± 0.5 kPa), were
used to monitor the hydrological response during WD cycles. Five TDR-310s and five WP-257s were inserted into the soil
column from the two opposite sides of the plexiglass column, respectively, with the same height spacing of 10 cm from top
to bottom.
Drainage measurement device, including two electronic balances, were used to record the cumulative water drainage from
the soil column.

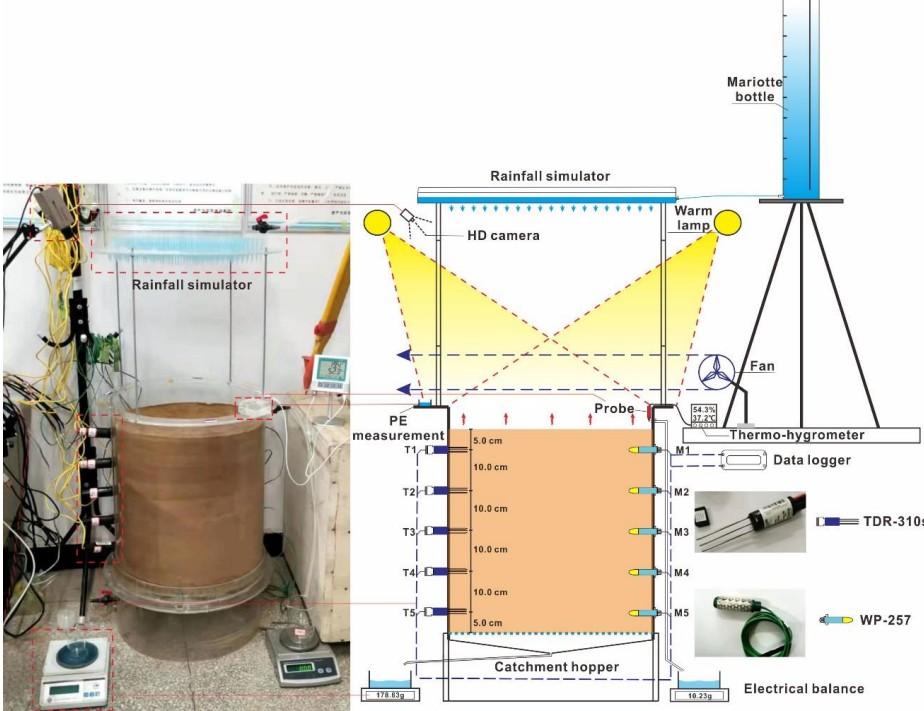


**Fig. 1** Schematic design and photos of the soil column test
2.2 Materials
The soil used in the test was taken from Zongyang county Anhui, China. Table 1 shows the basic physical parameters and
main mineral composition of the soil samples. The soil found in this study is classified as weak expansive soil. The saturated
hydraulic conductivity was measured on reconstituted soil cores with a dry density of 1.55g/cm³ (the same as the soil column).
In addition, the shrinkage curve of the saturated soil core was also obtained using a similar method proposed in Wen et al.
(2021). The difference is that we measured the vertical deformation in regular time intervals instead of continuous monitoring.
Fig. 2 shows the variation of soil porosity with the volumetric water content.
**Table 1** Basic physical parameters of the soil sample

| Gs (-) | $\omega_{opt}$ | $\rho_{d,max}$ | $L_l$ (%) | $P_l$ (%) | $\delta_{ef}$ (%) | $C_{Illite}$ | $C_{Kaolinite}$ | $C_{Quartz}$ | $C_{Albite}$ | $K_s$ |
|--------|--------|--------|-------|-------|-------|-------|-------|-------|-------|-------|
| 2.73 | 0.17 | 1.7 | 38.7 | 18.9 | 42.7 | 43-57 | 4-12 | 34-47 | 0-11 | $8.3\times10^{-7}$-$1.3\times10^{-6}$ |

Gs - specific gravity (-);

$\omega_{opt}$ - optimal moisture content (g/g); $\rho_{d,max}$ – the maximum dry density (g/cm³);

$L_l$ - liquid limit (%); $P_l$ - Plastic limit (%);

$\delta_{ef}$ - Free swelling ratio (%);

$C_{Illite}$, $C_{kaolinite}$, $C_{Quartz}$ and $C_{Albite}$ – content of illite, kaolinite, quartz and albite, respectively (%);

$K_s$ – Saturated hydraulic conductivity (m/s)

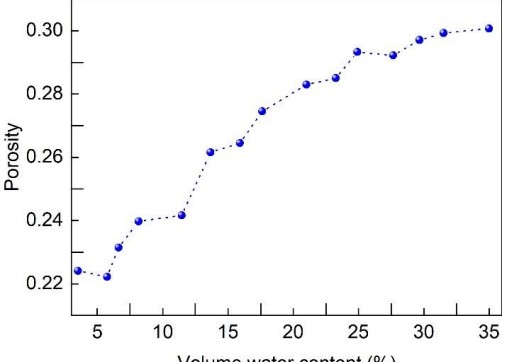


**Fig. 2** Shrinkage curve of the test soil
To ensure the homogeneity of the soil column, soil samples were compacted in 10 layers, and each layer was 5 cm thick. Prior
to filling soil into the plexiglass column, the soil samples with the total weight required for each layer were prepared according
to the designed density (dry density of 1.55g/cm³) and gravimetric water content (10%). Then, the soil samples were
compacted in the plexiglass column using a rubber hammer. The soil column was constructed within one day. After that, the
soil column was allowed to stand for 3 days to obtain stable records of the hydrological sensors.

2.3 Data collection
In the soil column test, the following data was collected:
(1) Boundary conditions: rainfall intensity ($r$, mm/h), potential evaporation ($PE$, mm/h) at 1 h time interval, temperature ($T$, ℃)
and relative humidity ($RH$, %) at 5 min time interval.
(2) Hydrological data: volume water content ($\theta_{exp}$, %) and soil matrix suction ($S_{exp}$, kPa) in different depths at 5 min time
interval, cumulative drainage from the top ($D_{top}$, g) and bottom ($D_{bottom}$, g) of the soil column.
(3) Crack geometric data: crack ratio ($w_{c,exp}$), crack aperture ($w_{j,exp}$), and the maximum crack depth ($d_{max}$, mm). The $w_{c,exp}$
and $w_{j,exp}$ were obtained via processing the crack photos which were taken at 20 min intervals during drying periods. The
image processing method mainly includes two steps as shown in Fig. 3. The $d_{max}$ was measured by thin wire before each
rainfall event.

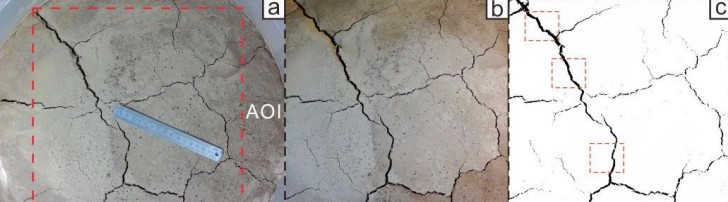


Fig. 3 Process of crack image processing. (a) a photo obtained from the HD camera, 800 pixels × 1400 pixels; (b) crack
image after cropping and pixel enhancement, 1044 pixels × 1005 pixels; (c) crack image after binarization and denoise, and
the crack ratio was calculated as the crack area divided by the overall AOI area, the crack aperture was calculated as the
average value of crack aperture from three different positions.

2.4 Test procedure
The overall experimental process included two stages of WD cycles. The purpose of the first stage was to generate a relatively
stable surface pattern of the desiccation cracks. It started from 2022/01/05 15:00 to 2022/02/28 9:00, including thirteen WD
cycles.
The second stage started from 2022/02/28 9:00 to 2022/03/28 22:30, including seven WD cycles. Fig. 4 presents the variation
of rainfall, evaporation, temperature and relative humidity in the entire experiment process. Because the two warm lamps and
fan were closed during the night, two kinds of evaporation intensity can be observed during the drying periods. In addition,
the average environment temperature in the 5$^{th}$ WD cycle was higher because we turned up the power of the two warm lamps.
In this current study, we mainly focus on the second stage of WD cycles.

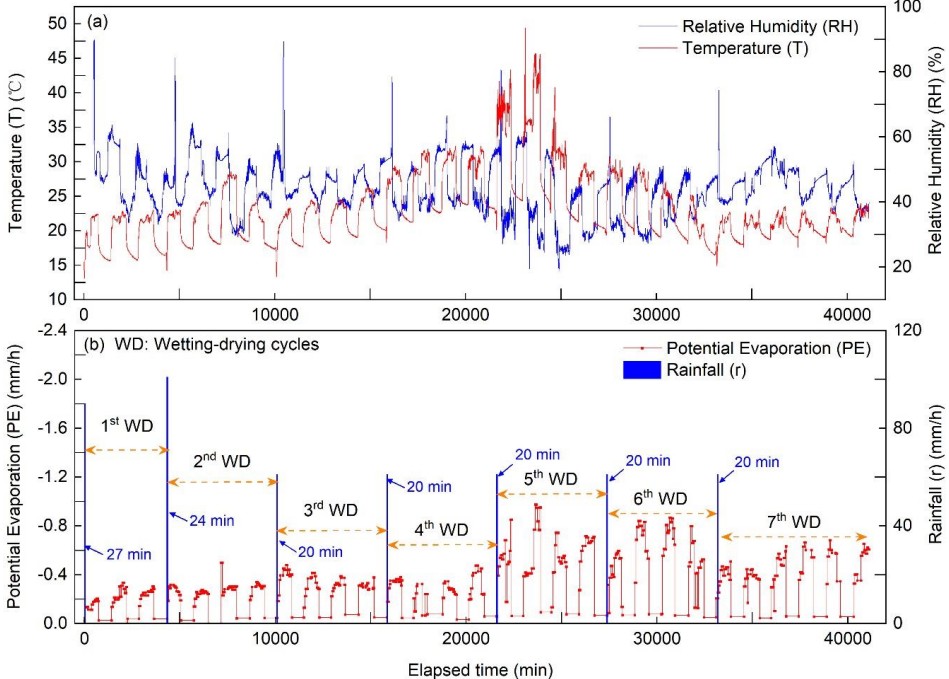


**Fig. 4** Environmental conditions of the experiment. (a) time series of temperature and relative humidity; (b) rainfall intensity
and potential evaporation.

**3.   Model Description**
3.1 Dual-permeability model (DPM)
The DPM concept used in this study corresponds to the one developed by Gerke and Van Genuchten (1993a). The model
divides the flow domain into two overlapping and interacting continua according to the volumetric ratios of each domain,
where two coupled 2-D Richards' equations are used to describe the matrix flow and preferential flow as
$$C_c(h)\frac{\partial h_c}{\partial t} = \nabla[K_c(h)\nabla(h_c + z)] - \frac{\Gamma_w}{w_c} \quad (1)$$
$$C_m(h)\frac{\partial h_m}{\partial t} = \nabla[K_m(h)\nabla(h_m + z)] + \frac{\Gamma_w}{w_m} \quad (2)$$
$$\Gamma_w = \alpha_w K_a(h_c - h_m) \quad (3)$$
$$w_c + w_m = 1 \quad (4)$$
Where
subscript "$c$" and "$m$" indicate the crack and matrix domains, respectively;
$h$ (m) is the pressure head;
$C$ represents the specific water capacity, $d\theta/dh$ (1/m);





$\theta$ (-) is the volumetric water content;
$K$ (m/s) is the isotropic hydraulic conductivity;
$z$ (m) is the elevation head;
$w$ (-) is the volumetric ratio of the crack domain or matrix domain over the bulk soil volume;
$\Gamma_w$ is the water exchange term (1/s) between the two domains;
$\alpha_w$ (1/m$^2$) is the effective water transfer coefficient;
$K_a$ (m/s) is the interface hydraulic conductivity.
The hydraulic properties of the two domains are parameterized based on the Mualem-van Genuchten soil-water retention
curves (SWRC) (Mualem, 1976; Van Genuchten, 1980) as
$S_e(h) = \dfrac{\theta - \theta_r}{\theta_s - \theta_r} = \left[ 1 + (|\alpha h|)^n \right]^{-m}$     (5)
$K(S_e) = K_s K_r(S_e) = K_s S_e^{0.5} [1 - (1 - S_e^{1/m})^m]^2$     (6)
where $S_e$ (-) is the effective saturation; $\theta_s$ (-) and $\theta_r$ (-) are the saturated and residual volumetric water content, respectively;
$\alpha$ (1/m), $n$ (-) and $m$ (-) are fitting parameters; $K_s$ (m/s) is the saturated hydraulic conductivity;. $K_r$ (-) is the relative
hydraulic conductivity.
According to Gerke and Van Genuchten (1993a), the total porosity $\varepsilon$ (-), total volume water content $\theta$ (-), total hydraulic
conductivity $K$ (m/s) and total volumetric flux (m/s) in terms of the volume ratio of each domain can be expressed as
$\varepsilon = w_c \varepsilon_c + w_m \varepsilon_m$     (7)
$\theta = w_c \theta_c + w_m \theta_m$     (8)
$K = w_c K_c + w_m K_m$     (9)
Note that the total porosity $\varepsilon$ is define as the total pore volume ($V_p$) divided by total soil volume ($V$), while $\varepsilon_m$ ( or $\varepsilon_c$ ) is
defined as the pore volume in matrix ($V_{p,m}$) (or crack, $V_{p,c}$) domain divided by the volume of that domain ($V_m$ or $V_c$). The total
volume water content has the same definition.
In the case of a DPM model, specified flux $i$ is divided between the matrix and crack domains as
$i = w_c i_c + w_m i_m$     (10)
where $i_c$ and $i_m$ are the effective boundary fluxes into each domain (m/s).
Considering a rainfall condition, the effective boundary fluxes of the two domains are initially equal to rainfall intensity ($r$)
due to the infiltration capacity of each domain is larger than $r$ (Dusek et al., 2008), and therefore the boundary fluxes of each
domain can be written as
$i_c = r$     (11)
$i_m = r$     (12)
As the soil keeps wetting, the decrease of the pressure head gradient may firstly lead to the infiltration capacity of matrix
domain dropping to a value less than $r$. Then, ponding occurs on the surface of the soil matrix and the boundary condition
changes to a specified pressure head boundary. This transformation can be achieved in COMSOL using a combined type of
boundary (Dirichlet and Neumann) proposed by Chui and Freyberg (2009). Once ponding occurs on the matrix domain, the
surplus water from that domain infiltrates into the crack domain and its effective flux increases to
$i_c = (r - w_m i_m) / w_c$     (13)
when the retained water volume in the cracks exceeds its storage capacity, water will pond on the surface of the crack domain.
Considering an evaporation condition, the Wilson-Fredlund-Barbour-Penman experimental function model (Wilson et al.,
1997) was used to calculate the actual evaporation of each domain



$AE/PE = \exp\left(\dfrac{-Sg\,\omega_v}{\xi(1-h_a)\gamma_w R(T_s + 273.15)}\right)$    (14)
Where
AE is the actual evaporation;
PE is the potential evaporation measured in the experiment;
$S$ (kPa) is total matric suction at the soil surface;
$g$ (m/s$^2$) is the gravitational acceleration constant;
$\omega_v$ is molecular mass of water, 0.018kg/mol;
$\xi$ is a dimensional empirical parameter with a suggested value of 0.7;
$h_a$ is relative humidity of overlying air;
$\gamma_w$ is unit mass of water, 9.807 kN/m$^3$;
R is universal gas constant, 8.314J/(mol·K);
$T_s$ (°C) is the soil surface temperature.

3.2 Dynamic dual-permeability model (DPMDy)
3.2.1 Porosity description
In Stewart et al. (2016a); (2016b) and Stewart (2018), the total porosity ($\phi_{max}$) of a cracked soil was divided into three domains:
aggregates (or soil matrix), cracks (voids from horizontal deformation induced by desiccation cracks) and subsidence (voids
from vertical deformation induced by desiccation cracks). In Stewart et al. (2016a), the distributions of these domains change
as a function of a unified water content, $U$
$\phi_{max} = \phi_{matrix}(U) + \phi_{crack}(U) + \phi_{sub}(U)$    (15)
where the subscripts matrix, crack and sub refer to the aforementioned three domains. In this study, we assume that the
horizontal deformation dominates the formation of desiccation cracks, thus $\phi_{sub}(U)$ can be neglected.
Stewart et al. (2016a) then deduced the porosities of each domain as:
$\phi_{matrix}(U)=(\phi_{max} - \phi_{min})(\dfrac{p+1}{p+U^{-q}}) + \phi_{min}$    (16)
$\phi_{crack}(U)=(\phi_{max} - \phi_{min})(\dfrac{1-U^q}{1+pU^q})$    (17)
where $p$ and $q$ are functional shape parameters; $\phi_{max}$ is the maximum porosity of a soil core prior to shrinkage and thus also
represents the total porosity; $\phi_{min}$ is the minimum porosity of the matrix domain; $U$ is a unified water content (defined as
water content $u$ divided by its saturated value $u_{max}$), which can be approximately estimated to be the saturation degree ($S_{e,m}$)
in an SWRC function of the soil matrix (Stewart et al., 2016a). Indeed, Eq. (16) represents a shrinkage curve function in which
four parameters can be obtained through a shrinkage test.
Substituting $S_{e,m}$ as $U$ and incorporating Eq. (5) into Eq. (16) and Eq. (17), we can obtain the porosity of the two domains as
a function of pressure head $h$
$\phi_{matrix}(h)=(\phi_{max}-\phi_{min})(\dfrac{p+1}{p+S_{e,m}^{-q}})+\phi_{min} = (\phi_{max}-\phi_{min})\left(\dfrac{p+1}{p+\left(\left[1+(|\alpha_m h_m|)^{n_m}\right]^{-m_m}\right)^{-q}}\right)+\phi_{min}$    (18)
$\phi_{crack}(h)=(\phi_{max}-\phi_{min})(\dfrac{1-S_{e,m}^q}{1+pS_{e,m}^q}) = (\phi_{max}-\phi_{min})\left(\dfrac{1-\left(\left[1+(|\alpha_m h_m|)^{n_m}\right]^{-m_m}\right)^q}{1+p\left(\left[1+(|\alpha_m h_m|)^{n_m}\right]^{-m_m}\right)^q}\right)$    (19)





With these porosity equations in mind, we can rewrite Eq. (4) and Eq. (7) as:
$\phi_{\max}=w_c\varepsilon_c+(1-w_c)\varepsilon_m$    (20)
Because the crack domain is mainly composed of voids, we here assume that $V_{p,c}$ equals to $V_c$, and thus $\varepsilon_c=1$. Through this
assumption, we obtained a physically-consistent definition of how the porosity and crack volume vary as functions of
saturation degree as follow
$w_c\varepsilon_c=w_c=\phi_{crack}(S_{e,m})=(\phi_{\max}-\phi_{\min})(\dfrac{1-S_{e,m}{}^q}{1+pS_{e,m}{}^q})$    (21)
$\varepsilon_m=\dfrac{\phi_{matrix}(S_{e,m})}{1-w_c}=\left[(\phi_{\max}-\phi_{\min})(\dfrac{p+1}{p+S_{e,m}{}^{-q}})+\phi_{\min}\right]/\left[1-(\phi_{\max}-\phi_{\min})(\dfrac{1-S_{e,m}{}^q}{1+pS_{e,m}{}^q})\right]$    (22)

3.2.2 Water content and hydraulic conductivity
In terms of Eq. (8), the total water content of the soil volume can be expressed as:
$\theta=\phi_{crack}(h)\theta_c+(1-\phi_{crack}(h))\theta_m$    (23)
Regarding the hydraulic conductivity of each domain, the classical DPM often assumed it equals to the product of a fixed $K_s$
and the relative hydraulic conductivity of the corresponding domain. The following equations are obtained according to Eq.

275    (6).

$K_m=K_{m,s}K_r(S_{e,m})=K_{m,s}S_{e,m}{}^{0.5}[1-(1-S_{e,m}{}^{1/m_m})^{m_m}]^2$    (24)
$K_c=K_{c,s}K_r(S_{e,c})=K_{c,s}S_{e,c}{}^{0.5}[1-(1-S_{e,c}{}^{1/m_c})^{m_c}]^2$    (25)
where $K_{c,s}$ and $K_{m,s}$ refer to the saturated hydraulic conductivity in crack and matrix domains, respectively.
However, the $K_{c,s}$ and $K_{m,s}$ are transient variables that changes with the crack geometries in crack domain and porosity in
matrix domain, which should be taken into consideration in a shrinking-swelling soil. To solve this issue, Stewart et al. (2016b)
further deduced models that describe the relationships between $K_{m,s}$, $K_{c,s}$ and $S_{e,m}$.
$K_{m,s}(S_{e,m})=K_{m,\max}\left(\dfrac{p+1}{p+S_{e,m}{}^{-q}}\right)$    (26)
$K_{c,s}(S_{e,m})=K_{c,\max}\left(\dfrac{1-U^q}{1+pS_{e,m}{}^q}\right)^2$    (27)
where $K_{c,\max}$ is the maximum saturated hydraulic conductivity of the crack domain (at $S_{e,m}=0$) when the crack aperture
achieves the maximum value; $K_{m,\max}$ is the maximum saturated hydraulic conductivity of the matrix domain (at $S_{e,m}=1$)
when the radius of cylindrical pores in that domain achieves the maximum value (See Eq. (25) and Eq. (27) in Stewart et al.
(2016b)). In the DPMDy model, we here set $K_r(S_{e,c})$ to 1 in Eq. (25). This modification means that the magnitude of $K_c$
only depends on the crack area or the saturated degree of the soil matrix domain.
$K_c=K_{c,s}=K_{c,s}(S_{e,m})=K_{c,\max}\left(\dfrac{1-U^q}{1+pS_{e,m}{}^q}\right)^2$    (28)
Incorporating Eq. (26) and Eq. (27) into Eq. (9) obtains:





$$K_s = \phi_{crack}(h)K_{c,\max}\left(\frac{1-S_{e,m}{}^q}{1+pS_{e,m}{}^q}\right)^2 + (1-\phi_{crack}(h))K_{m,\max}\left(\frac{p+1}{p+S_{e,m}{}^{-q}}\right) \quad (29)$$
Note that $K_{m,\max}$ can be obtained by laboratory-based infiltration test through a saturated soil core prior to shrinkage. Then,
Eq. (29) can be used to fit the $K_{c,\max}$ through the overall saturated hydraulic conductivity (measured $K_s$) under different
crack volume or ratio. Alternatively, $K_{c,\max}$ can also be approximately calculated as
$$K_{c,\max} = \frac{w_{j,\max}{}^2 g}{12v} \quad (30)$$
where $w_{j,\max}$ stands for the maximum crack aperture measured in experiment (m), g is the gravity acceleration constant
(m/s$^2$), and $v$ is the water kinematic viscosity (m$^2$/s). This equation is a relation to the cubes of the aperture of a crack with
respect to the crack inner flux, which is based on the derivation of laminar flow between parallel plates for Hagen-Poiseuille
type of flow (Snow, 1965).
Eventually, we can simulate the hydrological process with considering the dynamic changes of desiccation cracks by
incorporating Eq. (19), Eq. (21), Eq. (26), Eq. (27) and Eq. (28) into the DPM.

**4. Experimental results**
4.1 Crack dynamic changes
Fig 5 presents typical images of crack evolution during each WD cycle. Intuitively, it seems that the crack area and width did
not show an obvious increasing trend with the WD cycles as expected. Conversely, during the 1st to 4th WD cycles, the cracks
at the same moment after rainfall (Fig 5b2-4) and the final state (Fig 5c2-4) decreased significantly even though the
environmental temperature ($T$) and the potential evaporation ($PE$) increased in these periods. The cracks increased
significantly since the 5th WD cycle, but most of them were finer than before. Overall, cracks in the 1st WD cycle are wider
than those formed in other cycles.

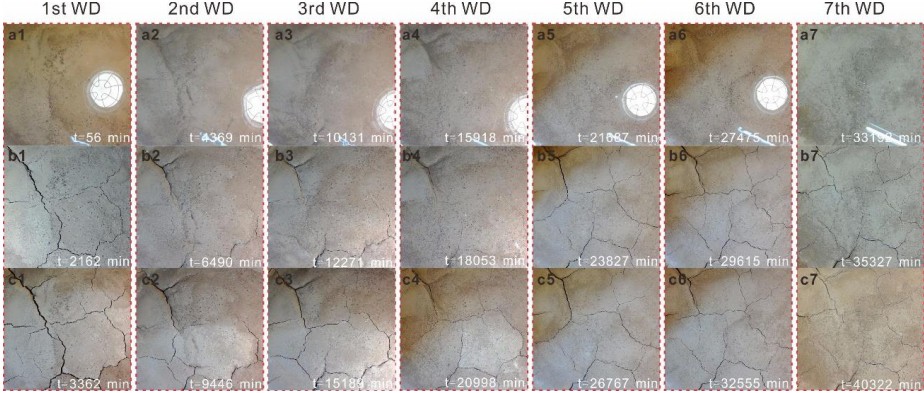

**Fig. 5** Typical images of crack evolution in seven wetting-drying cycles. (a1-7) water ponds on the soil surface after rainfall;
(b1-7) crack images at the 2135th min after each rainfall; (c1-7) crack images at the end of the final high evaporation period
during each wetting-drying cycle

Fig 6 quantitatively shows the variation of crack ratio ($w_{c,exp}$) and crack aperture ($w_{j,exp}$) in the experiment. Overall, the
variation curves corresponded to the intuitive descriptions mentioned above. Especially, an unexpected result was that the $T$





and *PE* in the 5[th] and 6[th] WD cycles were higher than in previous cycles, but their maximum $w_{c,exp}$ and $w_{j,exp}$ became
smaller. During a single WD cycle, the $w_{c,exp}$ and $w_{j,exp}$ have a similar trend, which shows a dramatic decrease during
rainfall, rapid increase in high evaporation periods and slow increase or even decrease in low evaporation periods. More
specifically, during the rainfall periods, the crack closure process was not significant until the water ponded on the soil matrix,
then ponded water flowed into the cracks, leading to acceleration of the crack closure. Note that cracks were not completely
closed even when they were full of water (Fig 5a1-7). The minimum crack ratio under such conditions is approximately 0.1%.
In the evaporation periods, the maximum crack ratio reaches 2.87% and the maximum crack aperture reaches 2.6 mm. In
addition, Fig 7 shows the maximum crack depth ($d_{max}$) measured after each cycle. It can be seen that $d_{max}$ increased
substantially after the 1[st] WD cycle and then slightly increased in the last six cycles, with a maximum value of 23.8 cm.

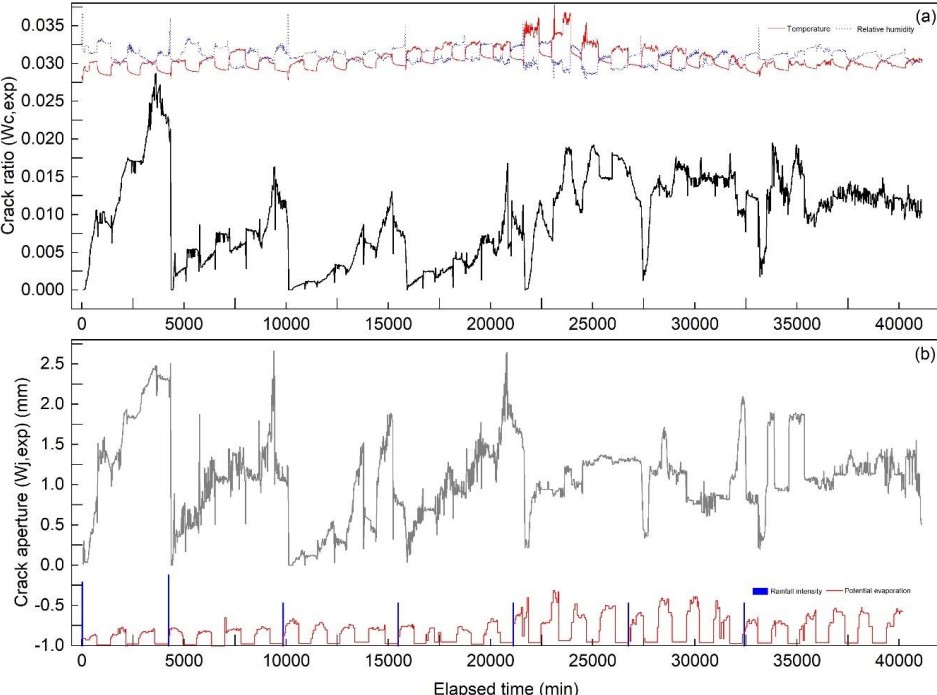


**Fig. 6** Time series of crack geometries. (a) crack ratio; (b) crack aperture

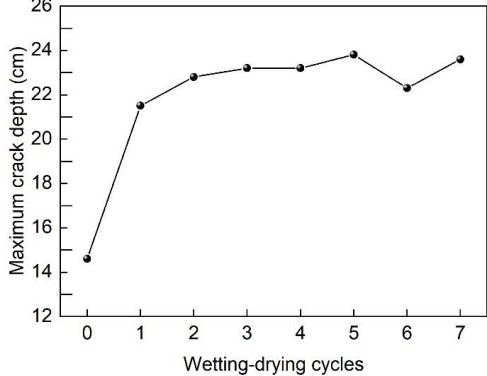


**Fig. 7** The maximum crack depth measured after each wetting-drying cycle

4.2 Hydrological response
Table 2 presents the manually recorded results of external hydrological responses involving ponding and drainage during each





WD cycle. It can be seen that the ponding occurred on the soil surface within 5 min after each rainfall. The ponding duration
in each rainfall mainly decreased with WD cycles. Note that the ponding depth in each rainfall was below the upper drainage
outlet. Regarding the water drainage, approximately 1.4 kg of water (the total water mass was 8 kg) was leaked during the 1st
rainfall due to the interspace between the soil and the plexiglass column and the hydrological sensors. Then, we sealed the
interspace using clay powder and polyurethane cement (soft materials without constrain effects on the soil swelling) after
each drying process, and subsequently, no water drainage was observed at the bottom outlet.
**Table 2** Statistical results of external hydrological responses

| Wetting-drying cycles | 1st | 2nd | 3rd | 4th | 5th | 6th | 7th |
|---|---|---|---|---|---|---|---|
| $t_p$ (min) | 4.1 | 1.8 | 1.2 | 1.2 | 1.2 | 2.2 | 2.8 |
| Ponding duration (min) | 70 | 160 | 68 | 47 | 34 | 25 | 23 |
| Drainage (g) | 1412 | - | - | - | - | - | - |
| *$t_p$ (min) – beginning of ponding after each rainfall | | | | | | | |

Fig 8 shows the internal hydrological responses recorded by the soil moisture and water potential sensors. Because the M2
and M4 were damaged during soil compaction, no matric suction data was obtained at their depths. Overall, water content at
all depths increased during rainfall and decreased during evaporation, where T1 showed the most sensitive responses to the
WD cycles. During rainfall, the time for water content to respond to each rainfall increased with depths, but the time difference
among all depths decreased significantly since the 2nd WD cycle. During the drying periods, an interesting phenomenon was
that the water content at 5 cm depth showed an overall decline trend, but transient increases of water content frequently
appeared during low evaporation periods. Such transient increases seem to be related to the slow decrease of crack ratio as
mentioned in section 4.1. Regarding the matric suction, its variation trend was similar to the water content but showed more
delayed responses to the environmental conditions, especially in the last three WD cycles. Additionally, Fig 8b also implies
that soil at 5 cm depth reached saturation during each rainfall, while soil below the 25 cm depth was in the unsaturated state
in the whole experiment process.

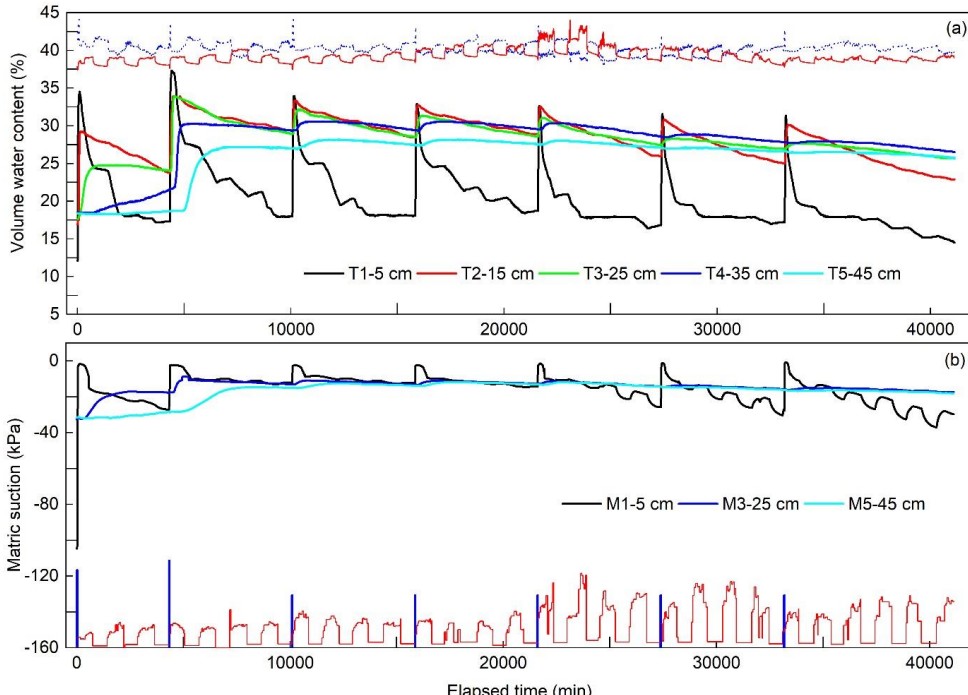


**Fig.8** Time series of volume water content (a) and matric suction (b) at different depths.

## 5. Numerical simulation

5.1 Set-up of numerical model

The single-domain model (SDM), dual-permeability model (DPM) and dynamic DPM (DPMDy) were implemented in a finite

element solver for Richards' equation as part of the COMSOL Multiphysics software (Comsol 5.6). As shown in Fig 9, they

have the same 2-D size, boundary conditions, mesh structure and initial condition. The model domain is 0.5 m by 0.5 m, same

as the soil column. Because the measured maximum crack depth was 23.8 cm, we specified the crack domain existing within

the upper 25 cm depth of the soil column.

The boundary conditions at the top were set as combined type of boundary conditions (as mentioned in section 3.1) for

representing the rainfall, ponding and evaporation process recorded in the experiment; the bottom side is a seepage boundary

condition; the left and right sides of the model are no-flux boundaries.

Because the pressure head in the surface area may change frequently and drastically during WD cycles, a refined mesh

structure with dense boundary layers was used to capture the transient hydrological conditions. The boundary layers included

15 layers of rectangular grid, with the minimum and maximum thick of approximately 0.04 cm and 0.3 cm, respectively. A

coarser free-triangle mesh (average length of 1.8 cm) was defined below the boundary layers. The initial condition both in

matrix and crack domains was set as the distribution of pore water pressure measured from the experiment prior to the 1$^{st}$ WD

cycle.

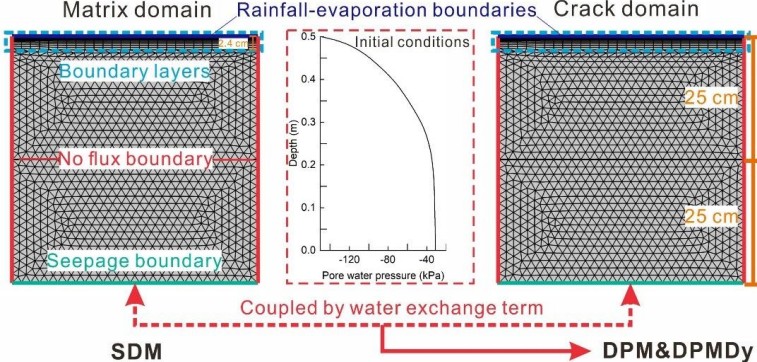

**Fig. 9** Set-up of the 2-D numerical model for the SDM, DPM and DPMDy

5.2 Parameters

5.2.1 Shrinkage parameters

As shown in Fig 10, using Eq. (18) to fit the measured shrinkage curve in Fig. 2, we obtained the four shrinkage parameters

as $\phi_{min}=0.22$, $\phi_{max}=0.30$, $p=8.8\pm4.84$, $q=2.71\pm0.85$. Then, the variation of porosity in crack domain (or crack ratio

$w_c$) and matrix domain ($\varepsilon_m$) could be obtained using Eq. (21) and Eq. (22), respectively. Note that the minimum $w_c$ calculated

by Eq. (21) was set as 0.001 considering the incomplete closure of cracks during rainfall.





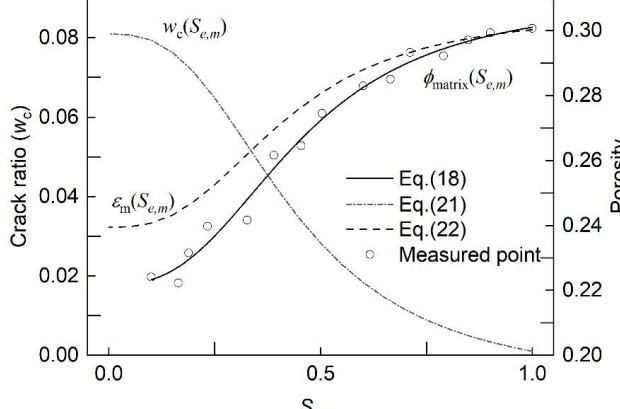


**Fig. 10** Fitted shrinkage curve (solid line) and modeled porosity variation of matrix (dash line) and crack domains (dash-dot

line)

5.2.2 Soil water retention parameters
**Fig 11** shows the measured matric suction versus volume water content (or measured SWRC) at different depths. It can be
seen that the WD cycles lead to hysteretic curves in the SWRC at 5 cm and 25 cm depths, while that at the 45 cm depth rarely
show hysteretic curves. This result may also indicate that most of the cracks exist within the upper 25 cm depth of the soil
column. In this study, we simply estimated an approximate single SWRC of the soil matrix through experiment data instead
of incorporating the hysteretic curves into the model. For instance, the estimated SWRC curve in **Fig 11a** lies between the
wetting SWRC and drying SWRC to capture the overall characteristics of wetting-drying SWRC as far as possible. Note that
the shape parameter $n$ in the upper matrix domain is slightly smaller than the lower one considering the upper soil matrix may
become denser after long-time WD cycles (13 times, 54 days). Regarding the SWRC of the crack domain with macropore-
dominated space, the SWRC parameters of that domain were set with a greater saturated water content ($\theta_{c,s}$= 0.99), a lower
value of air entry pressure ($\alpha$ = 1.5) and a steeper slope ($n_c$ = 2) than that of the matrix domain.

5.2.3 Hydraulic conductivity
As mentioned in Eq. (29), the maximum saturated hydraulic conductivity of matrix domain ($K_{m,\max}$) equals the saturated
hydraulic conductivity ($K_s$) measured in laboratory. Here, we set $K_{m,\max}$ = 1.16×10$^{-6}$ m/s. Regarding the $K_{c,\max}$, it was
calculated using Eq. (30), where the $w_{j,\max}$ was set to 2.6 mm obtained from Fig 6b. Then, the variation curve of transient
saturated hydraulic conductivity of the matrix domain ($K_{m,s}$) and the crack domain ($K_{c,s}$) could be obtained using Eq. (27)
and Eq. (28), respectively. Note that here we slightly modified Eq. (28) as follow.
$$K_{c,s}(S_{e,m})=K_{c,\max}\left(\frac{1-S_{e,m}{}^q}{1+pS_{e,m}{}^q}\right)^2+K_{c,\min} \quad (28\text{-}b)$$
This modification not only avoided the $K_{c,s}$ dropping to zero thus benefits the numerical convergence, but also was
reasonable when considering the incomplete closure of cracks during rainfall. The $K_{c,\min}$ was also estimated using Eq. (30)
with a suggested $w_{j,\max}$ = 0.01 mm. Further, the variation of $K_m$ and $K_c$ with the pressure head ($h$) in the DPMDy could
be calculated by combining Eq. (24), Eq. (26) and Eq. (28). **Fig 12** presents $K_m$ and $K_c$ in the three models. Note that the
pressure head in $K_c(h_m)$ of the DPMDy refers to that of the matrix domain ($h_m$), while $h$ in $K_c(h_c)$ of the DPM refers to
that of the crack domain ($h_c$).



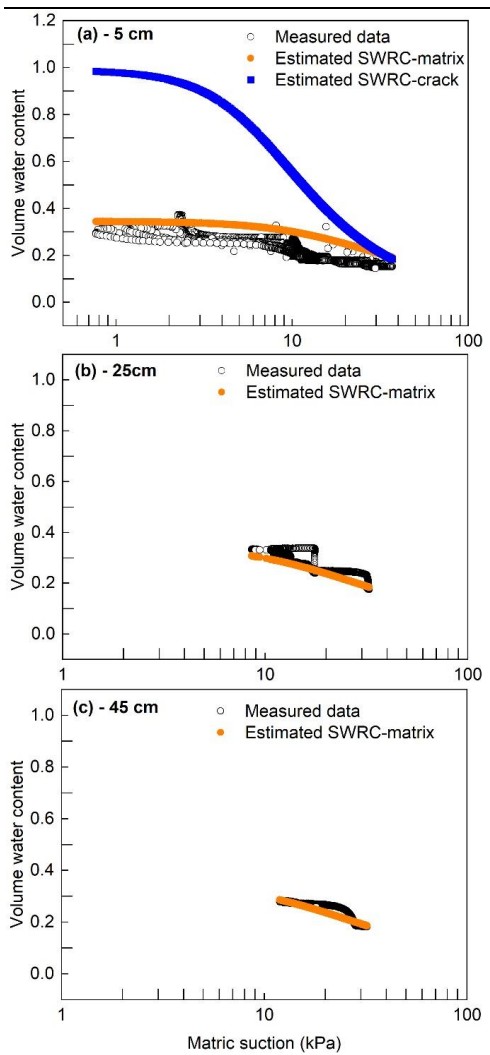


**Fig. 11** Measured and estimated SWRC at different depths. (a) 5 cm; (b) 25 cm; (c) 45 cm






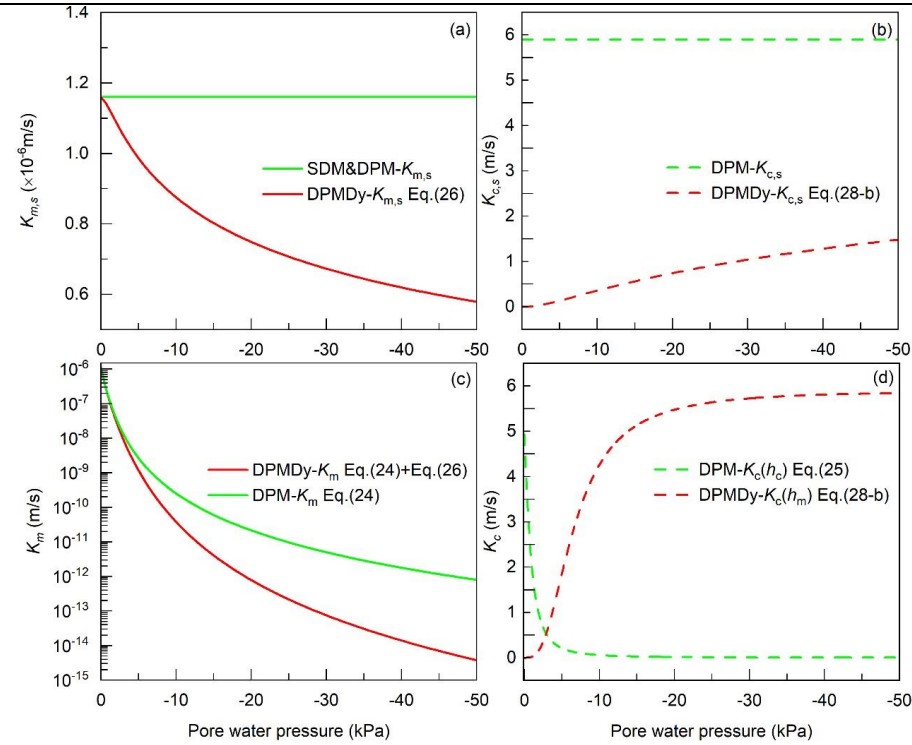

**Fig. 12** Modeled hydraulic conductivity of each domain in the three models. (a) Saturated hydraulic conductivity of the matrix domain; (b) saturated hydraulic conductivity of the crack domain; (c) transient hydraulic conductivity of the matrix domain; (d) transient hydraulic conductivity of the crack domain

In the dual-permeability concept, another important parameter is the hydraulic conductivity of the interface between matrix and crack domains ($K_a$). Generally, $K_a$ was often estimated as the arithmetic mean of hydraulic conductivity of the two domains (Arora et al., 2011; Coppola et al., 2012; 2015; Gerke and Van Genuchten, 1993b; Laine-Kaulio et al., 2014; Shao et al., 2015). However, this approximation may overestimate the $K_a$ when the hydraulic conductivity of the crack domain is much higher than that of the matrix domain, especially in cracked clays. In our current study, a $K_a$ function reformulated by (Gerke et al., 2013) was adopted.

$$K_{a_{min}} = \begin{cases} \min\{K_m(h_c), K_c(h_c)\} & h_c \geq h_m \\ \min\{K_m(h_m), K_c(h_m)\} & h_c < h_m \end{cases} \quad (31)$$

This formulation represents that the flow occurs from the highest head toward the lowest head but regulated by the less permeable of the two subsystems in that instant of time (Aguilar‐López et al., 2020).

Regarding the $\alpha_w$, experimental results presented by Song et al. (2018) showed that the saturated $K_a$ may be 1 order of magnitude larger than the $K_{m,s}$ which will represent an enlarging coefficient ranging from 10 to 18. Hence, the $\alpha_w$ was set as 10 m$^{-2}$ considering the saturated $K_{a_{min}}$ determined by Eq. (31) equals to the $K_{m,s}$.

All parameters for the SDM, DPM and DPMDy are listed in **Table 3**.

**Table 3** Summary of parameters for the SDM, DPM and DPMDy

| Model | Symbol | Parameter name | Units | Upper layer | Lower layer |
|---|---|---|---|---|---|
| SDM | $\theta_{m,s}$ | Saturated water content of matrix domain | (-) | 0.345 | 0.345 |
| DPM | $\theta_{m,r}$ | Residual water content of matrix domain | (-) | 0.01 | 0.01 |





| | | | | | |
|---|---|---|---|---|---|
| DPMDy | $\alpha_m$ | Mualem-van Genuchten fitting parameter of matrix domain | (1/m) | 0.6 | 0.6 |
| | $n_m$ | Mualem-van Genuchten fitting parameter of matrix domain | (-) | 1.65 | 1.8 |
| | $K_{m,\max}$ | The maximum $K_s$ of matrix domain before shrinkage | (m/s) | $1.16 \times 10^{-6}$ | $1.16 \times 10^{-6}$ |
| DPM DPMDy | $\theta_{c,s}$ | Saturated water content of crack domain | (-) | 0.99 | - |
| | $\theta_{c,r}$ | Residual water content of crack domain | (-) | 0.01 | - |
| | $\alpha_c$ | Mualem-van Genuchten fitting parameter of crack domain | (1/m) | 1.5 | - |
| | $n_c$ | Mualem-van Genuchten fitting parameter of crack domain | (-) | 2 | - |
| | $K_{c,\max}$ | The maximum $K_s$ of crack domain | (m/s) | 5.9 | - |
| | $K_a$ | Hydraulic conductivity of the interface | (m/s) | $K_{amin}$ | - |
| | $a_w$ | Mass transfer coefficient | (1/m²) | 10 | - |
| DPMDy | $\phi_{\max}$ | The maximum porosity of a soil core before shrinkage | (-) | 0.3 | - |
| | $\phi_{\min}$ | The minimum porosity of a soil core after shrinkage | (-) | 0.22 | - |
| | p | Shape parameter of soil shrinkage curve in Eq. (18) | (-) | 10 | - |
| | q | Shape parameter of soil shrinkage curve in Eq. (18) | (-) | 3.5 | - |
| DPM | $w_c$ | Constant crack ratio using in DPM | (-) | 0.01; 0.03 | - |
| * SDM: single-domain model; DPM: dual-permeability model neglecting crack dynamic changes; DPMDy: Dynamic DPM; $w_c$ = 0.01 and 0.03 refers to the average and the maximum value of the measured crack ratio, respectively. | | | | | |


5.3 Simulation results
5.3.1 Boundary flow
**Fig. 13** shows the temporal evolution of the boundary flow velocity simulated by the SDM, DPM and DPMDy. As shown in
**Fig. 13a1 and a3**, during drying periods, the matrix domain dominates the soil evaporation process and was responsible for
97%-99% of the total evaporation in all the dual-permeability models. The matrix evaporation rate ($e_m$) simulated by the
DPMDy was overall lower than that of the SDM and DPM during high-intensity evaporation periods, but the crack
evaporation rate ($e_c$) simulated by the DPMDy, especially during the last three drying periods, was approximately one to two
orders of magnitude larger than that of the DPM (see the enlarged image in Fig. a1).
With regard to the wetting process, **Fig. 13a2 and a4** represent two typical infiltration patterns before and after the 5th drying
period (with significantly increased evaporation intensity). Overall, matrix flow still dominated the infiltration process in all
the dual-permeability models due to the relatively small crack ratio and depth. For the SDM, all the rainfall infiltrates into the
soil during the beginning of rainfall events. When the soil surface gets saturated, water ponding occured and the soil infiltration
rate gradually decreased. In the DPM and DPMDy, the surplus water after matrix ponding infiltrates into the crack domain as
preferential flow, and water will pond on the overall soil surface when the crack domain reached its storage capacity. Recall
that the crack volume in the DPMDy decreases with the matrix getting moist, while that in the DPM keeps constant.
Consequently, the ponding time of the crack domain simulated by the DPMDy in the 3nd rainfall event (inflection point of the
red dash line in Fig. 13a2) was 1.6 and 4.8 min earlier than that of the DPM-0.01 and DPM-0.03, respectively. The cumulative
preferential flow simulated by the DPMDy was 87.4% and 95.2 % less than that of the DPM-0.01 and DPM-0.03, respectively.
Similar rainfall pattern was obtained during the 6th rainfall event.



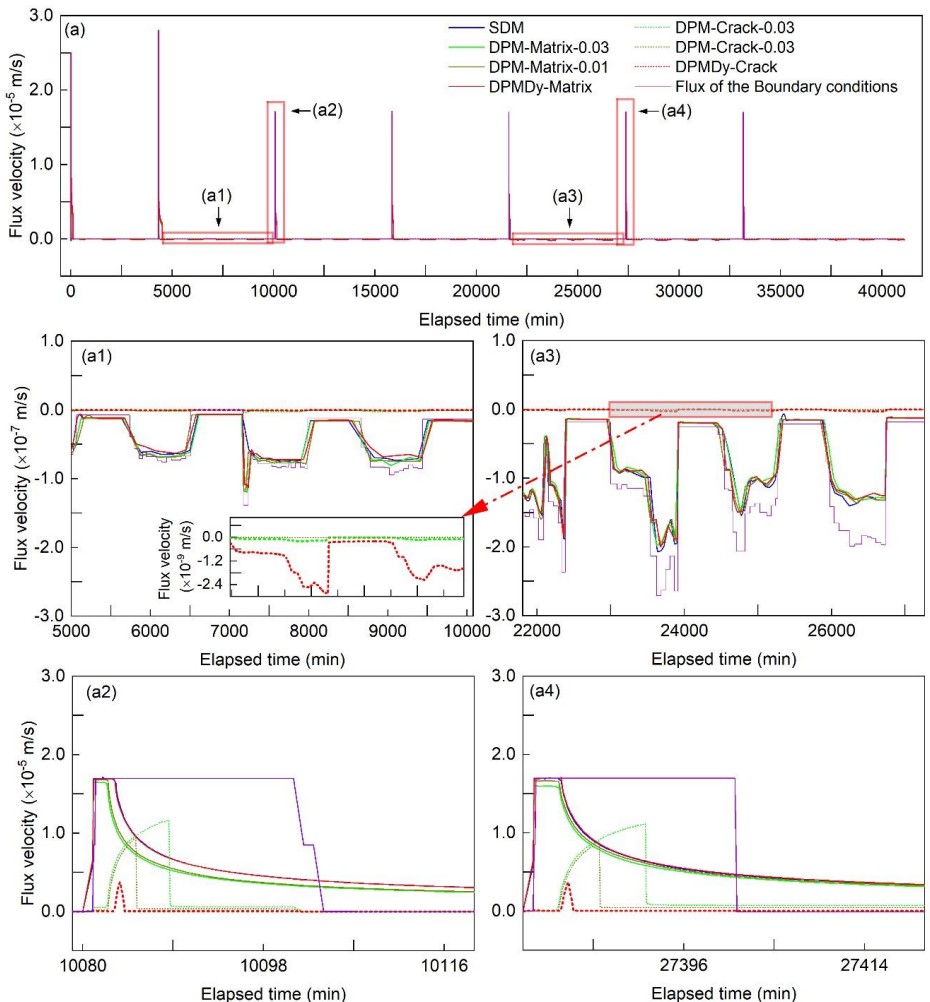


**Fig. 13** Boundary flow simulated by the SDM, DPM and DPMDy. (a) Flow velocity of the boundary conditions and simulated
results; (a1) and (a2) are the enlarged images of the flow velocity during the 2st drying and 3nd wetting process, respectively;
(a3) and (a4) are the enlarged images of the flow velocity during the 5th drying and 6th wetting process, respectively. The
positive value is for infiltration and negative for evaporation.

5.3.2 Water balance
By integrating the boundary flow velocity in Fig. 13a, the total cumulative flux for the experiment and the three models were
obtained (**Fig 14a)**. In the experiment, the variation of water flux was estimated by calculating the sum of the difference
between $\theta_{ini}$ (initial volume water content) and $\theta_{t=i}$ (volume water content at any time) in the five monitoring depths.
Meanwhile, the water evaporation during water ponding was also estimated and added to the total flux volume. Regarding
the numerical model, the water balance was obtained by integrating all flow components along the upper and lower boundaries.
The steep increase stage of each curve represents cumulative input water flux during wetting periods and the gradual decrease
stage represents cumulative output water flux during drying periods. To evaluate the performance of each model on the water
balance, the measured cumulative input and output water fluxes in each wetting and drying stage were compared to the
simulated ones (Fig. 14b).
In Fig. 14a, the results show that the total infiltration ($I_{t,inf}$) and evaporation flux ($E_{t,eva}$) estimated from measured $\theta_{exp}$ were





171 mm and 138.95 mm, respectively. The $I_{t,inf}$ was 5.86 % less than the supplied water (183.44 mm) due to the water leakage.
The $E_{t,eva}$ was 16.48 % less than the cumulative $PE$ (166.36 mm) because of the limit of the soil actual evaporation. Regarding
the simulation results, the coefficient of determination ($R^2$) and intercept were used to evaluate the errors made by the three
models. As shown in Fig. 14b, the slope of each fitting curve was fixed as 1. The SDM and DPMDy have relatively smaller
intercepts and slightly higher $R^2$ than that of the DPM-0.01 and DPM-0.03, indicative of a better coincidence to the measured
data. Overall, the errors in water balance caused by the three models were acceptable in this study.

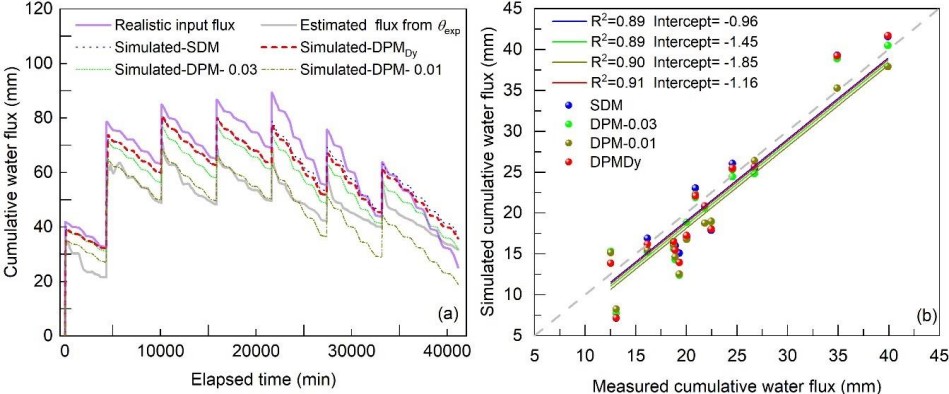


**Fig. 14**. Water balance for the measured and simulated results (a) Temporal evolution of total water flux calculated from the
measured water content, SDM, DPM and DPMDy; (b) measured versus modeled cumulative flux during each drying and
wetting stage

5.3.3 Crack dynamic changes and hydrological response
**Fig. 15** shows part of the comparison results between the measured data and the three models. Detailed descriptions of all the
comparison results are presented in **Appendix A**. Overall, all models show similar response trends with the measured data.
Divergences among the three models mainly appeared during drying.
In **Fig. 15a**, the simulated surficial $w_{c,sim}$ was not only generally close to the $w_{c,exp}$ in value and trend, but also it captured
the transient slow decrease of $w_{c,exp}$ during low evaporation periods. Notably, significant overprediction appeared in the 6[th]
and 7[th] wetting-drying cycles.
In **Fig. 15b**, the matric suction ($S_{sim}$) at the 25 cm depth simulated by SDM and DPMDy was close to each other and had an
average divergence 2.26 kPa to the measured data. The $S_{sim}$ simulated by DPM had a greater average divergence of 3.4 kPa
to the measured data. They showed systematic underprediction compared to the $S_{sim}$ simulated by SDM and DPMDy, but
their differences became smaller with the increasing WD cycles.
In **Fig. 15c**, the total volumetric water content $\theta_{sim}$ simulated by SDM was much lower with respect to the DPMDy and
DPM. The $\theta_{sim}$ simulated by DPM-0.01 and DPM-0.03 overpredicted the volumetric water content. The DPMDy provided
better prediction results but also showed slight underprediction to the measured data at the last two WD cycles.

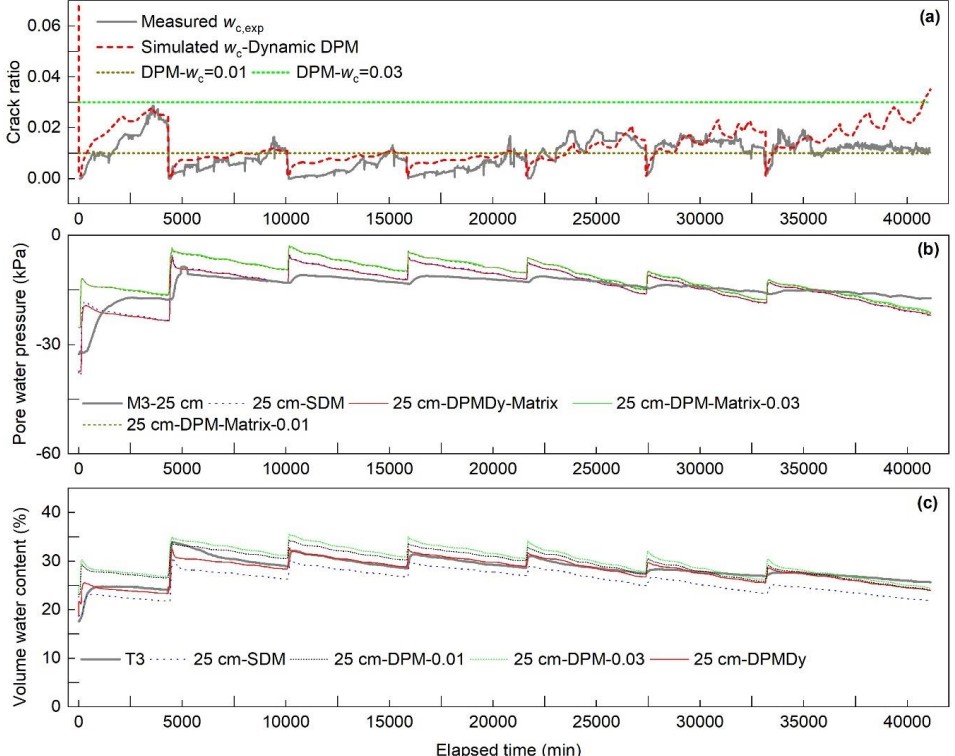

**Fig. 15** Temporal evolution of the measured and simulated crack ratio, matric suction and volumetric water content. (a) Measured and simulated crack ratio on soil surface; (b) Measured and simulated matric suction at 25 cm depth; (c) Measured and simulated total water content at 25 cm depth

## 6. Discussions

6.1 Crack dynamic changes

Our experimental results demonstrated that the crack evolution is not always positively correlated to the increase of the WD cycles, $T$ and $PE$. For instance, the 5 cm $\theta_{exp}$ at the end of the final three WD cycles was lower than that in the 1$^{st}$ WD cycle due to the increased $T$ and $PE$, but the maximum $w_{c,exp}$ measured during the final three WD cycles was much less than that in the 1$^{st}$ WD cycle. From the energy-driven perspective, soil cracking and propagation can be regarded as a process that the shrinkage energy (or stress), built up from the evaporation and thermal radiation, was released until a critical moment when the tensile strength of soil is reached (Peron et al., 2009). If the environmental condition changes in a stable range, the desiccation cracks will vary within the crack pattern and the maximum $w_{j,exp}$ that were formed under the maximum shrinkage energy. In this case, new desiccation cracks will not appear in the remained soil matrix during WD cycles (**Fig 5b1-b4**). One reason is that the shrinkage energy can be fully released via previous cracks. The other reason is that the shrinkage energy is not high enough to split the soil matrix that has a denser structure (or higher tensile strength) than its initial state prior to shrinkage (Luo et al., 2021). However, once the evaporation rate and thermal radiation increase to exceed the stable range, higher shrinkage energy will lead to new cracks appearing in the soil matrix that will concurrently restrain the width increase of the previous cracks (Wang et al., 2018). This is the reason that cracks in the final three WD cycles are finer than the first four WD cycles. Our model describes the crack evolution mainly from the hydrological-driven perspective that assumed the surface crack pattern has become stable after undergoing 13 WD cycles and has a constant function relationship with the water content. Indeed, this assumption is reasonable for natural soils under atmospheric environmental conditions. However, our experiment not only used reconstituted soil but also intensely changed the environmental conditions since the

515 $5^{th}$ WD cycle. Therefore, the model overpredicted $w_{c,exp}$ at the end of the $6^{th}$ and $7^{th}$ WD cycles.

516 In addition, another interesting phenomenon is the transient decrease of $w_{c,exp}$ and increase of 5 cm $\theta_{exp}$ during low

517 evaporation periods, which we called as 'self-closure' process. In light of **Fig 6** and **Fig 8**, the self-closure process appeared

518 always accompanied by relatively high $RH$. From the insight of the experiment, it is natural and common to infer that the

519 moist air wetted the surface soil from top to bottom, resulting in the self-closure phenomenon. Interestingly, our model does

520 not incorporate the vapor flow into the boundary conditions, and also the evaporation boundary only involves the outflow of

521 water, but it still managed to captured the self-closure process. **Fig 16** shows the crack images at t = 3702 and 4327 min as

522 well as the corresponding cloud chart of $\theta_{sim}$. It can be seen that the soil surface became moist during the low evaporation

523 period, which is a typical external phenomenon reflecting the self-closure process. The simulation results show that $\theta_{sim}$

524 near the surface soil increased during evaporation while $\theta_{sim}$ at deep soils decreased, indicative of evaporation inducing the

525 deep water move up and wet the surface soil from bottom to top. We further found that the process occurred because the water

526 flow driven by the soil water potential gradients, existing between the wet and dry soil layers, overcame the gravity. Indeed,

527 this kind of 'hydraulic lift' process frequently occurs in planted soils where root zone soil can force water flow from moist

528 deep soil layers to dry shallow soil layers (Richards and Caldwell, 1987; Bauerle et al., 2008), but was rarely reported in

529 homogeneous bare soil. We infer that the evaporation boundary conditions using Eq. (14) might play a positive role in leading

530 water move up and constraining it within the surficial soil depths when the evaporation intensity decreased. In any case, our

531 results provide an additional possible explanation to the self-closure phenomenon. Further quantitative analysis based on gas-

532 liquid two phase flow model is needed to compare the contribution of 'hydraulic lift' and moist air to the self-closure process

533 of cracks.

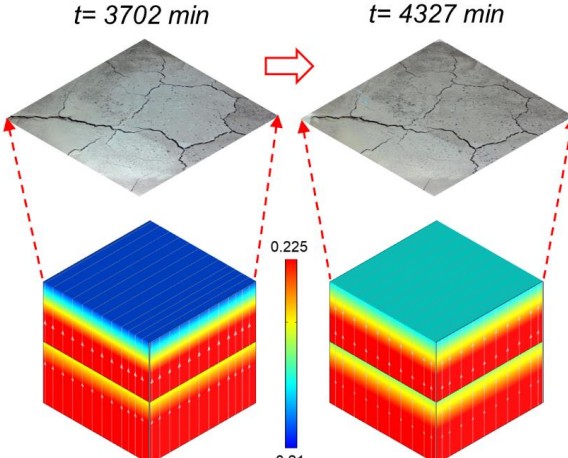

534

535 **Fig. 16** Self-closure process of cracks captured in experiment (Upper figures) and numerical model (Lower figures) during

536 the low evaporation process. The left part is at the beginning of the final low evaporation stage during the $1^{st}$ drying periods,

537 while the right part is at the end of the final low evaporation stage during the $1^{st}$ drying periods.

538

539 6.2 Water flow with dynamic changes of desiccation cracks

540 6.2.1 Water fluxes

541 As mentioned in section 5.3.1, during the drying process, the matrix and crack evaporation simulated by the DPMDy are

542 overall lower and higher than other models, respectively. It can be explained by looking at the variation of boundary $K_m$ and

543 $K_c$ in each model. Take the time span in **Fig.13a2** as an example, because the DPMDy considers the effects of matrix shrinkage

544 on the $K_m$ using Eq. (26), the $K_{m,DPMDy}$ is always approximately 20% and 30% lower than that of the SDM and DPM,



respectively **(Fig. 17a)**. On the contrary, because the DPM links the $K_c$ with the saturation degree of the crack domain (see
Eq. (25)), the $K_{c,DPM}$ is destined to decrease with the decreased saturation degree of the crack domain induced by drying, while
the $K_{c,DPMDy}$ increases with the crack development induced by drying in light of Eq. (28-b). The ultimate $K_{c,DPMDy}$ is 80%
higher than the $K_{c,DPM}$ **(Fig. 17b)**. Indeed, the decrease of $K_c$ with the drying process is an unrealistic and physically-
unreasonable results. We can image that after long-term drought, the $K_{c,DPM}$ will decline to nearly zero according to Fig. 12d,
which will greatly underestimate the propagation of the PF-DC in the subsequent storm event. However, many laboratory and
field experiments have observed that heavy rainfall following a long-term drought facilitated PF-DC (Baram et al., 2012a;
2013; Greve et al., 2010; Kurtzman and Scanlon, 2011; Schlögl et al., 2022). By contrast, the DPMDy has the potential to
capture this process for its increasing $K_c$ with the enlarging desiccation crack during the long-term drought. In this study,
because the experiment scale (or crack volume) is small, the increment of PF-DC simulated by the DPMDy after high-intensity
evaporation is not significant (despite increment = 25%), but we believe the DPMDy will have a better performance in a
larger scale (i.e slope scale).

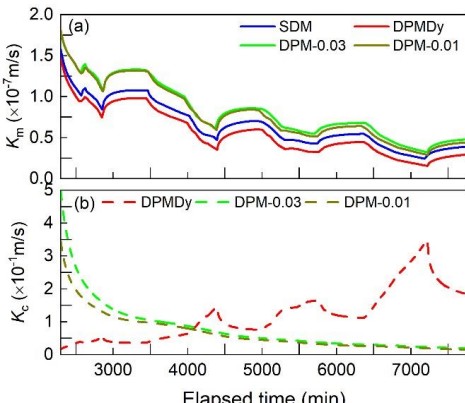


**Fig. 17** Variation of boundary $K_m$ and $K_c$ in each model during the 5th drying periods. (a) $K_m$; (b) $K_c$

6.2.2 Water exchange and distribution
For the dual-permeability model, the two domains are coupled by the water exchange term (Eq. 3) that is governed by the
pressure head difference between the two domains ($\Delta h = h_c - h_m$), water exchange coefficient ($\alpha_w$) and the hydraulic
conductivity between the two domains ($K_a$). The higher the $\Gamma_w$, the quicker the two domains equilibrate. Generally, the
higher $\Gamma_w$ leads to faster water exchange from the crack domain into the matrix domain and thus boosts the contribution of
preferential flow on the water distribution in the soil matrix. According to the previous studies, the commonly used magnitude
of the product of saturated $\alpha_w K_a$ in clay soils ranges from $10^{-5}$ m⁻¹s⁻¹(Aguilar‐López et al., 2020) to $10^{-6}$ m⁻¹s⁻¹ (Coppola
et al., 2012; 2015; Gerke and Maximilian Köhne, 2004; Vogel et al., 2000). In this study, the saturated $\alpha_w K_a$ is $1.16 \times 10^{-5}$
m⁻¹s⁻¹, which falls in the reasonable range. Building on the above statement, the $\Delta h$ and water exchange rates ($\Gamma_w / w_m$) for
both the DPM and DPMDy at the 5 cm, 15 cm and 25 cm depths during the 6th rainfall event are graphed in **Fig. 18**.
As shown in **Fig. 18a1-a3**, $\Delta h$ at all depths simulated by both the DPM and DPMDy rapidly reaches a positive peak value
and gradually decreases with the rainfall process. The rapidly increasing positive value is because the crack domain gets
saturation earlier than the surrounding soil matrix due to the influx of preferential flow and the small crack storage space in
this study. The decrease of the $\Delta h$ is ascribed to the increase of $h_m$ with water exchanging from crack to matrix domain.
Notably, the crack closure process during rainfall process leads to decrease of crack volume (or crack water storage space),
the 'water table' (saturated zone) in the shrinking cracks elevates faster than that in the constant larger crack volume, which
means the $h_c$ simulated by DPMDy is higher than the DPM-0.01 and DPM-0.03. Consequently, the time for $\Delta h$ reaching
the peak value simulated by the DPMDy is the earliest at all the three depths, then followed by the DPM-0.01 and DPM-0.03.
The $\Gamma_w / w_m$ simulated by the DPMDy shows the similar trend to the $\Delta h$ (**Fig. 18b1-b3**). During the 6[th] rainfall event, its
cumulative $\Gamma_w / w_m$ at the 5 cm, 15 cm and 25 cm depths is (26%, 50%), (10%, 26%) and (3%, 14%) larger than that of the
DPM-0.01 and DPM-0.03, respectively.
This result means that the crack closure during wetting benefits the building-up process of the pressure head in the crack
domain and thus can promote water exchange from crack into matrix domain. It corresponds to some experimental results
that the PF-DC also exists and leads water rapidly infiltrate into soils even desiccation cracks are nearly closed (Baram et al.,
2012a; Greve et al., 2010; Luo et al., 2021; Sander and Gerke, 2007; Tuong et al., 1996). It also means using DPM may
overestimate the flux of PF-DC, but underestimate the water exchange coming from the PF-DC. Because the experimental
scale, crack ratio and depth in this study is small, the difference of simulation result involving the matric suction and water
content between the DPM and DPMDy is not very significant. However, we can image that the deviation caused by the DPM
at a larger scale will be more significant, especially in a typical shrinking-swelling soil slope under long-term WD cycles.

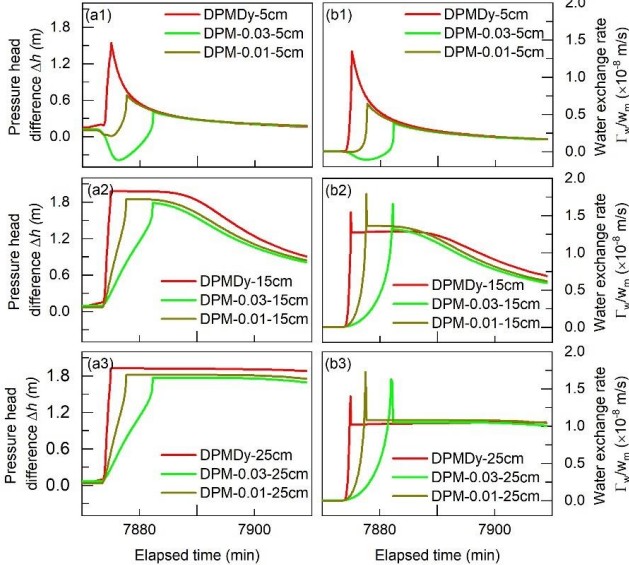


**Fig. 18** Pressure head difference (a1-a3) and water exchange rate (b1-b3) between the two domains at the 5 cm, 15 cm and
25 cm depths during the 6[th] rainfall event. The positive value of water exchange rate is for the water flowing from the crack
to the matrix domain, while the negative value for the opposite direction

6.3 Model performance
In this study, the simulation results show that the DPMDy, which incorporates the dynamic changes of desiccation cracks and
hydraulic conductivity into the dual-permeability model, has an overall better performance than the SDM and DPM. With
regard to the water flux, while the three models all have acceptable errors to the measured data, the DPM overpredicted the
water flux of PF-DC but underestimate the water exchange from cracks to soil matrix. It implies that adopting a constant
crack volume in the DPM model, whether it is an average or a maximum value of the measured crack ratio, will overestimate
the PF-DC, which may be unsuitable to evaluate the irrigation efficiency. With regard to the matric suction (or pore water
pressure), although the SDM has good performance as the DPMDy does, it significantly underpredicted the volume water





content and thus may overestimate landslide stability in a moisture-content-dependent threshold method. Further, we expect
that the SDM may show much poorer performance if one applies it to scenarios where the cracks are deeper and the soil has
a higher swelling-shrinking ability than that of our experiment. A comprehensive model sensitivity analysis will be conducted
in our future work.
Compared to other dynamic preferential flow models, the DPMDy developed in this study also has its unique advantages.
Firstly, the variation of crack volume (or crack ratio) in our model is deduced from the changes of matrix porosity due to
shrinkage and thus has a physically-consistent as well as a universal definition. Instead, Coppola et al. (2012); (2015) linked
the crack ratio to the suction head with an empirical natural logarithm function, which not only implies a disconnection
between hydrological properties and porosity in the crack domain but also may not be universal when applying it to other
kinds of soils. Secondly, a common defect both in Coppola et al. (2012); (2015) and classical DPM is that they often set the
hydraulic conductivity of the crack domain ($K_c$) varies as a function of the saturated degree calculated from the SWRC of the
crack domain (i.e Eq. (25)). This will lead to an unreasonable extremely low $K_c$ in drying initial conditions (Aguilar‐López
et al., 2020). In our model, we set the relative hydraulic conductivity of the crack domain to unit ($K_r =1$). It ensures that the
magnitude of $K_c$ only depends on the crack area or the saturated degree of the soil matrix domain, which provides a potential
solution for remedying the shortcoming mentioned above. Thirdly, compared to some dynamic preferential flow models
neglecting the water exchange between the two domains (Jamalinia et al., 2020; Kroes et al., 2000; Luo et al., 2021; Stewart,
2018) or adopting an improper exchange term (Coppola et al., 2012; 2015), our model tentatively adopts an improved
exchange term proposed by Gerke et al. (2013), which is proved to be a logically correct and satisfactory improvement in
simulating water exchange in our experiment.
Because our model neglects the effect of hysteresis both in the soil deformation and soil-water retention curve, it inevitably
caused some errors when compared to the measured water content, especially for the surficial soil layer that has been
significantly affected by the WD cycles. Our future work will try to incorporate the hysteresis effect into the current model to
further improve the prediction strength. Besides, we have to remind again that because the shrinking-swelling model in our
method is developed based on the hydrological-driven perspective, it may be more suitable in the natural soil layer where the
crack pattern has a stable state after long-term WD cycles.

## 7. Conclusions

This study combined an experimental study and a numerical simulation to quantify the preferential flow induced by dynamic
changes of desiccation cracks (PF-DC). A soil column infiltration test under wetting-drying conditions was conducted to
investigate dynamic changes of desiccation cracks and the accompanying water infiltration process. The variation of crack
geometry, including crack ratio, width and depth were measured. The soil volumetric water content, matric suction and water
drainage were also monitored. A new dynamic dual-permeability model (DPMDy) was developed to account for the PF-DC,
which includes physically-consistent functions in describing the variation of both porosity and hydraulic conductivity in crack
and matrix domains. The performance of the single-domain model (SDM), rigid dual-permeability model (DPM) and DPMDy
was evaluated by comparing their simulation results to the monitoring data.
Overall, the DPMDy performed not only better prediction on the crack evolution and hydrological response with respect to
the SDM and DPM, but also provided much better descriptions on the underlying physics involving the PF-DC. During the
drying periods, the matrix evaporation modeled by the DPMDy is lower than that of the SDM and DPM due to considering
the permeability decay induced by soil shrinkage. But the crack evaporation modelled in the DPMDy approach is the highest
because it managed to capture the raised crack permeability induced by drying-enlarging desiccation cracks. Compared to the
DPM with fixed crack volume, the DPMDy revealed that the crack closure process during wetting will lead to a faster pressure
head building-up process in the crack domain and higher water exchange rates from the crack to the matrix domain.





Additionally, using a fixed crack ratio in the DPM, whether it is the maximum or the average value from the experiment data,
will overestimate the infiltration fluxes of PF-DC but underestimate its contribution to the matrix domain.
The DPMDy developed here has a physically-consistent definition. It remedies the shortcomings of the RDPM and other
dynamic preferential flow models in defining the dynamic changes of desiccation cracks and hydraulic properties of the crack
domain and interface. Future works should focus on considering the hysteresis effect of the SWRC curve during wetting-
drying cycles in the model and its application to complex field situations.

**Appendix A**
**Fig. A1** and **Fig. A2** show the temporal evolution of the measured and simulated crack ratio on the soil surface, matric suction
(negative pore water pressure) and volumetric water contents at the five monitoring depths (5, 15, 25, 35 and 45 cm).
In **Fig A1a**, the simulated $w_{c,sim}$ was not only generally close to the $w_{c,exp}$ in value and trend, but also it captured the
transient slow decrease of $w_{c,exp}$ during low evaporation periods.
In **Fig A1b-f**, the matric suction ($S_{sim}$) simulated by SDM and DPMDy is close to each other and has average divergence of
2.75 kPa, 2.26 kPa and 5.02kPa to the measured data at the 5 cm, 25 cm and 45 cm depths, respectively. The $S_{sim}$ simulated
by DPM has a greater average divergence of 2.78 kPa, 3.4 kPa and 7.43 kPa to the measured data at the three corresponding
depths.
In **Fig A2a-e**, the volumetric water content $\theta_{sim}$ simulated by SDM was much lower than that simulated by DPMDy and
DPM. In most depths (except the 5 cm and 45 cm depth), SDM systematically underpredicted the volumetric water content
during both wetting and drying periods. By contrast, the $\theta_{sim}$ simulated by DPM-0.01 and DPM-0.03 overpredicted the
volumetric water content. The DPMDy gave overall better prediction results in most depths, but has significant divergences
to the measured data at the depth of 5 cm and so are the other two models.







**Fig. A1** Temporal evolution of the measured and simulated crack ratio and matric suction at different depths. (a) Measured and simulated crack ratio (Dynamic DPM) on soil surface; (b-f) Measured and simulated matric suction (Single domain model, DPM and Dynamic DPM) at depths of 5 cm, 15 cm, 25 cm, 35 cm and 45 cm.




**Fig. A2** Temporal evolution of the measured and simulated volumetric water content at depths of 5 cm, 15 cm, 25 cm, 35 cm
and 45 cm. Note that the simulated volumetric water content demonstrated here is the total volumetric water content that
combined with the combined matrix and crack domains using Eq. (8)
**Notation**

| | |
|---|---|
| PF-DC | Preferential flow induced by desiccation cracks |
| SDM | Single-domain model |
| EMs | Explicit crack models |
| DPoM | Dual-porosity model |
| DPM | Rigid dual-permeability model with fixed crack ratio and hydraulic conductivity |





| | |
|---|---|
| DPM-0.01 | Rigid dual-permeability model with crack ratio of 0.01 |
| DPM-0.03 | Rigid dual-permeability model with crack ratio of 0.03 |
| DPMDy | Dynamic DPM with changing crack ratio and hydraulic conductivity |
| WD cycles | Wetting-drying cycles |

| | |
|---|---|
| $\theta$ | Total water content (combined matrix and crack domains), $m^3m^{-3}$ |
| $\theta_{exp}$ | Volumetric water content measured in the experiment, $m^3m^{-3}$ |
| $\theta_m$ | Volumetric water content of the matrix domain, $m^3m^{-3}$ |
| $\theta_c$ | Volumetric water content of the crack domain, $m^3m^{-3}$ |
| $\theta_{m,s}$ | Saturated volumetric water content of the matrix domain, $m^3m^{-3}$ |
| $\theta_{m,r}$ | Residual volumetric water content of the matrix domain, $m^3m^{-3}$ |
| $\theta_{c,s}$ | Saturated volumetric water content of the crack domain, $m^3m^{-3}$ |
| $\theta_{c,r}$ | Residual volumetric water content of the crack domain, $m^3m^{-3}$ |
| $S_{e,m}$ | Saturation degree of the matrix domain, $m^3m^{-3}$ |
| $S_{e,c}$ | Saturation degree of the crack domain, $m^3m^{-3}$ |
| $\alpha_m$ | Parameter for the van Genuchten water retention curve of the matrix domain, 1/m |
| $n_m$ | Parameter for the van Genuchten water retention curve of the matrix domain, 1/m |
| $m_m$ | Parameter for the van Genuchten water retention curve of the matrix domain, 1/m |
| $\alpha_c$ | Parameter for the van Genuchten water retention curve of the crack domain, 1/m |
| $n_c$ | Parameter for the van Genuchten water retention curve of the crack domain, 1/m |
| $m_c$ | Parameter for the van Genuchten water retention curve of the crack domain, 1/m |
| $h_m$ | Pressure head of the matrix domain, m |
| $h_c$ | Pressure head of the crack domain, m |
| $C_c$ | Specific water capacity of the crack domain which is defined as $d\theta_c / dh_c$, 1/m |
| $C_m$ | Specific water capacity of the matrix domain which is defined as $d\theta_m / dh_m$, 1/m |
| $K_s$ | Total transient saturated hydraulic conductivity of the soil (combined matrix and crack domains), m/s |
| $K_c$ | Transient hydraulic conductivity of the crack domain, m/s |
| $K_{c,s}$ | Saturated hydraulic conductivity of the crack domain, m/s |
| $K_{c,max}$ | The maximum crack hydraulic conductivity when the crack reaches its maximum crack aperture, m/s |
| $K_{c,min}$ | The minimum crack hydraulic conductivity when the crack reaches its minimum crack aperture, m/s |
| $K_{c,r}$ | Relative hydraulic conductivity of the crack domain, $m^3m^{-3}$ |
| $K_m$ | Transient hydraulic conductivity of the matrix domain, m/s |
| $K_{m,s}$ | Saturated hydraulic conductivity of the matrix domain, m/s |
| $K_{m,max}$ | The maximum matrix hydraulic conductivity prior to soil shrinkage, m/s |
| $K_{m,r}$ | Relative hydraulic conductivity of the matrix domain, $m^3m^{-3}$ |
| $K_a$ | Hydraulic conductivity between the matrix and crack domains, m/s |
| $K_{a_{min}}$ | An improved hydraulic conductivity between the matrix and crack domains reformulated by Gerke et al. (2013), m/s |
| $\Gamma_w$ | Water exchange term between the crack and matrix domains, 1/s |
| $w_c$ | Crack ratio, which is defined as volumetric ratio between the crack domain and the overall soil volume, $m^3m^{-3}$ |
| $w_{c,exp}$ | Surface crack ratio measured in experiment, $m^2m^{-2}$ |
| $w_{j,exp}$ | Average crack aperture (or crack width) measured in the experiment, m |
| $w_{j,max}$ | The maximum average crack aperture measured in the experiment, m |
| $d_{max}$ | The maximum crack depth measured in the experiment, m |
| $w_m$ | Volumetric ratio between the matrix domain and the overall soil volume, $m^3m^{-3}$ |





| $\alpha_w$ | Effective water transfer coefficient, 1/m$^2$ |
| --- | --- |
| $V$ | Total soil volume (combined matrix and crack domains), m$^3$ |
| $V_m$ | Volume of the soil matrix domain, m$^3$ |
| $V_c$ | Volume of the crack domain, m$^3$ |
| $V_p$ | Total pore volume, m$^3$ |
| $V_{p,m}$ | Pore volume in the matrix domain, m$^3$ |
| $V_{p,c}$ | Pore volume in the crack domain, m$^3$ |
| $\varepsilon$ | Total soil porosity (combined matrix and crack domains), which is defined as $V_p/V$, m$^3$m$^{-3}$ |
| $\varepsilon_m$ | Effective porosity of the matrix domain, which is defined as $V_{p,m}/V_m$ |
| $\varepsilon_c$ | Effective porosity of the crack domain, which is defined as $V_{p,c}/V_c$ |
| $i$ | Total effective infiltration rate (combined matrix and crack domains), m/s |
| $i_m$ | Effective infiltration rate of the matrix domain, m/s |
| $i_c$ | Effective infiltration rate of the crack domain, m/s |
| $e_m$ | Effective evaporation rate of the matrix domain, m/s |
| $e_c$ | Effective evaporation rate of the crack domain, m/s |
| $r$ | Rainfall intensity, m/s |
| $AE$ | Actual evaporation rate, m/s |
| $PE$ | Potential evaporation rate, m/s |
| $S$ | Total matric suction at the soil surface, kPa |
| $S_{exp}$ | Soil matric suction measured in the experiment, kPa |
| $g$ | Gravitational acceleration constant, m/s$^2$ |
| $\omega_v$ | Molecular mass of water, kg/mol |
| $\xi$ | Dimensional empirical parameter with a suggested value of 0.7 |
| $h_a$ | Relative humidity of soil overlying air |
| $\gamma_w$ | Unit mass of water, kN/m$^3$ |
| R | Universal gas constant, J/mol·K |
| $T_s$ | Soil surface temperature, ℃ |
| $\phi_{max}$ | Total porosity (or the maximum porosity) of a soil core prior to soil shrinkage, which is defined as $V_p/V$ and thus equals to the $\varepsilon$, m$^3$m$^{-3}$ |
| $\phi_{min}$ | The minimum porosity of the matrix domain, m$^3$m$^{-3}$ |
| $\phi_{matrix}$ | Porosity of the matrix domain, which is defined as $V_{p,m}/V$, m$^3$m$^{-3}$ |
| $\phi_{crack}$ | Porosity of the crack domain, which is defined as $V_{p,c}/(V_m+V_c)$, m$^3$m$^{-3}$ |
| $\phi_{sub}$ | Porosity of the subsidence zone, which is defined as voids induced by soil subsidence divided by the total soil volume, m$^3$m$^{-3}$ |
| $U$ | A unified water content, which is defined as the gravimetric water content $u$ divided by its saturated value $u_{max}$ |
| $p$ | Functional shape parameters of the soil shrinkage curve |
| $q$ | Functional shape parameters of the soil shrinkage curve |
| $v$ | Water kinematic viscosity, m$^2$/s |
| $t_p$ | Beginning of ponding time after each rainfall, min |
| $\Delta h$ | Pressure difference between the crack and matrix domains, which is defined as $h_c - h_m$ |


**Code/Data availability**
The source code and the data generated from this study are available from the corresponding author upon reasonable request.

**Author contribution**





Yi Luo: Conceptualization, Methodology, Investigation, Writing-original draft preparation
Jiaming Zhang*: Supervision, Writing - review & editing, Project administration
Zhi Zhou: Resources, Software, Investigation
Juan P. Aguilar-Lopez: Writing - review & editing
Roberto Greco: Writing - review & editing
Thom Bogaard: Supervision, Writing - review & editing, Funding acquisition

**Competing interests**

Some authors are members of the editorial board of journal Hydrology and Earth System Sciences. The peer-review process was guided by an independent editor, and the authors have also no other competing interests to declare.


**Financial support**

This work was financially supported by the National Natural Science Foundation of China (42177166) and the Fundamental Research Funds for National University, China University of Geosciences (Wuhan). It was also partially funded by the Plan of Anhui Province Transport Technology Progress (grant 2018030) and Engineering Research Center of Rock-Soil Drilling & Excavation and Protection, Ministry of Education (202210).

694

**Acknowledgment**

This paper was written during visiting research exchange of Yi Luo at TUDelft in Summer 2022. The experiment was conducted from January to March 2022 in China, Professor Ming-jian Hu and his research group are thanked for their great help in providing TDR probes. Yi Luo's Chinese colleagues Yuhao Li, Zhan Yang, Xiang Li, Zijian Shen are thanked for their contribution on the experiment monitoring. The authors also would like to thank the editor and anonymous reviewers for their valuable comments that substantially improved this paper.






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
