# Peer review of "Effects of dynamic changes of desiccation cracks on preferential flow: Experimental investigation and"

_Hydrology and Earth System Sciences, 2022_

## Referee Comment (RC1)

**Effects of dynamic changes of desiccation cracks on preferential flow: Experimental investigation and numerical modeling**
by Luo et al.

In the paper, the authors deal with the important issue of preferential flow, which has been widely analysed in the literature of the last 30-40 years. The work carried out in the paper consists of including the swelling-shrinkage process, induced by changes in soil water contents, in the well-known Dual Permeability Model (DPM) proposed by Gerke and van Genuchten (1996), to account for the dynamic changes of fractures volume with soil wetting and drying.

The approach used by the authors incorporates the swelling-shrinkage approach proposed by Stewart et al. (2016a, b) to describe change of porosity in both the soil fracture and matrix domains.

The work is based on an important experimental work carried out in the laboratory on both small and relatively large columns filled with disturbed soil. The evolution of shrinkage with wetting and drying has been determined experimentally by analysing the images of the soil surface taken by a HD camera. Also, an improved exchange term proposed by Gerke et al. (2013) has been included to account for the exchange between the fracture and the matrix domains.

The data set coming from the column experiments has been used to evaluate the effectiveness of the simulations coming from the proposed model as compared to simulation coming from both a single domain and a rigid double domain model.

General remarks

Based on my reading of the manuscript, the paper is quite well structured. I found everything quite clearly written and explained. The issue dealt with is clearly discussed in the Introduction of the manuscript, with an exhaustive literature. The figures summarize quite clearly the results. Some parameters in the tables should be described better. Also, a table with some more information on the correlation among fitted parameters should be given, along with their confidence intervals. The materials and methods are well explained, with enough and clear details on the experimental work. The development of the fitting procedure is not completely clear. Results are complete and well-illustrated. Most of the discussion and conclusions coming from the numerical simulations seems well supported by the data.

To me, the issue dealt with in the paper is not novel. From a conceptual point of view, the paper mostly retraces the work already carried out by Coppola et al. (2012; 2015). Compared to the latter, the work under review incorporates a new approach for swelling-shrinkage changes of hydraulic properties in both the soil fractures and matrix, as proposed by Stewart et al. (2016a, b). Also, the soil considered is a reconstituted soil, differently from the work by Coppola et al., who calibrated, validated and tested the model on data coming from experiments involving in situ undisturbed soil plot and undisturbed soils samples taken from the soil matrix of the same plot.

Some statements in the Introduction and in the discussion and conclusions seems a bit forced and misleading. I will try to argue about them, also to discuss some other issues the authors dealt with in the manuscript.

In the Introduction, the authors state: "*Coppola et al. (2012); (2015) took another step forward to allowed crack volume and/or hydrological properties to vary as a function of soil shrinkage. However, the relationship proposed in the model, a natural logarithm function involving the suction head and crack proportion, lacks physical consistency with the variation of porosity. This implies a disconnection between hydrological properties and porosity in the crack domain.*" A similar statement may be found again in the conclusions of the paper under review. To me, this statement appears as a wrong and approximate interpretation and reproduction of a quite hurried conclusion drawn by Stewart et al. (2016a), who wrote (page

7912): "*Coppola et al. [2012, 2015] allowed β and/or the soil hydraulic properties (e.g., volumetric water content, hydraulic conductivity) to vary as a function of soil shrinkage. However, the relationships proposed in those models lack physical consistency, in that domain specific hydraulic properties (e.g., hydraulic conductivity) remain constant regardless of changes in porosity distribution (e.g., β). This disconnect (as they wrote in their paper) between hydraulic properties and swelling ….*".

As may be deduced in Stewart et al., the disconnection they speak about concerns the fact that hydraulic properties are not allowed to change with porosity changes. By reading carefully the paper by Coppola et al. (2012), this argument is unfounded. In the section 3.3, the authors clearly explained how the $\vartheta_a(h)$ and $K(\vartheta_a)$ ($\vartheta_a$ is the moisture ratio of the soil matrix) experimental data points measured on the soil cores were converted to as many $\theta_a(h)$ and $K(\theta_a)$ points by using the equation 10a and the $e_a(h)$ values measured at the same $h$. Thus, the $\theta_a(h)$ and $\theta_f(h)$ (and the corresponding $K(\theta_a)$ and $K(\theta_f)$) parameters comes from the $\vartheta_a(h)$ adjusted for the $e_a(h)$ data and, once used as input in the code, already account for the deformation of both domains with changing $h$. In other words, the $\theta$ and the $K$ values calculated during simulations for a given $h$ value at a given simulation node already accounts for the deformation of pore-size distributions of both the domains under swelling/shrinkage. What's more, the authors also allowed the fraction of the matrix and fracture porosity to change along with hydraulic properties. As this is not a simple task from an analytic point of view, they assumed a logarithmic function describing the ß(h) evolution, but this is another story and has nothing to do with the physical inconsistency and the disconnection between changes in porosity and hydraulic properties Stewart et al. discussed about.

As for the paper by Coppola et al. (2015), it simply showed three scenarios where the swelling-shrinking cycles were assumed to alternatively affect 1) only the hydraulic properties, 2) only the fraction of the two porosities with no effects on the hydraulic properties, 3) both, in the combined approach already presented in the 2012 paper. So, saying that these approaches do not account for changes in hydraulic properties is simply unfounded.

In any case, if the physical inconsistency lies in the disconnection between changes of porosity and corresponding changes in hydraulic properties, as argued by Stewart et al. (2016a, b), this could more apply to the paper under review. In fact, the authors should explain clearly where in their paper they change the hydraulic properties with swelling-shrinking cycles. If I well understood, their approach assumes fixed hydraulic property shape parameters (see table 3) for both the domains and the porosity is assumed to change according to equations 18 and 19. The Ks is scaled according to the changes in the porosity. I guess the saturated water content also scales similarly, even if I did not find any explanation of how the change in the porosity scales the water retention curves. Is Ks only a function of the porosity, rather than of the whole pore-size distribution? Do the authors believe that swelling-shrinkage simply scales the hydraulic properties (that is, swelling shrinkage has only effects on the total porosity of both the domains), as suggested by unchanged shape parameters of the soil hydraulic properties, or rather it changes the whole pore-size distribution (as considered by Coppola et al. in the 2012 paper and in the approach they called ßk in the 2015 paper)?

To me, it seems that the argument of the physical inconsistency was introduced in the paper rather to maintain the usefulness of using a swelling-shrinkage approach a bit different from that previously used. I find the Stewart et al. approach actually effective and physically attractive, but this do not requires arguing that the approach by Coppola et al. is physically inconsistent.

Other comments:
Page 4, line 134: This means that the authors assume the Ks and the hydraulic properties obtained on a reconstituted soil sample represent the Ks and the hydraulic properties of the

matrix in the reconstituted large soil column with the same bulk density. This is confirmed at page 14, line 397. This would imply that the Ks depends only on the bulk density of the soil. The authors should discuss more this point;

Table 1: I did not understand what is the optimal water content, ωopt, in the table;

Page 7, line 221. This is the first time you introduce COMSOL in the paper. I would explain here what it is.

Page 12, line 337. Did the authors account for this interspace when calculating the fractures volume change?

Table 3: as for the parameters pertaining to the DPM and DPMDy, are they the final parameters coming from fitting or are they the initial guess values? In any case, the fracture parameters seem quite strange. The saturated water content simply suggests a void without solid particles, which is not coherent with the n parameter, suggesting at least a sand porous medium;

Page 24, line 618. I did not understand the sentence "…improper exchange term (Coppola et al., 2012; 2015). Did these authors use an improper exchange term?

Section 6.3. Model performance: It would have been interesting to see a table in this section summarising quantitatively the effectiveness of the model fitting, with optimized parameters, correlation matrix, confidence intervals, …)

References

Gerke, H. H., and M. T. van Genuchten (1993a), A dual porosity model for simulating the preferential movement of water and solute in structured porous media, Water Resour. Res., 29, 305–319, doi:10.1029/92WR02339.

Coppola, A., Gerke, H. H., Comegna, A., Basile, A., and Comegna, V., 2012. Dual-permeability model for flow in shrinking soil with dominant horizontal deformation, Water Resour. Res., 48, 10.1029/2011wr011376, 2012.

Coppola, A., Comegna, A., Dragonetti, G., Gerke, H. H., and Basile, A., 2015. Simulated Preferential Water Flow and Solute Transport in 726 Shrinking Soils, Vadose Zone J., 14, 10.2136/vzj2015.02.0021, 2015.

Stewart, R. D., Rupp, D. E., Abou Najm, M. R., and Selker, J. S.: A Unified Model for Soil Shrinkage, Subsidence, and Cracking, 811 Vadose Zone J., 15, 1-15, 10.2136/vzj2015.11.0146, 2016a.

Stewart, R. D., M. R. Abou Najm, D. E. Rupp, and J. S. Selker, 2016a. Modeling multidomain hydraulic properties of shrink-swell soils, Water Resour. Res., 52, 7911–7930, doi:10.1002/2016WR019336

Gerke, H. H., Dusek, J., and Vogel, T., 2013. Solute Mass Transfer Effects in Two-Dimensional Dual-Permeability Modeling of Bromide Leaching From a Tile-Drained Field, Vadose Zone J., 12, 10.2136/vzj2012.0091, 2013.

---

## Author Comment (AC1)

**Reply to Professor Coppola's comments**

Dear Professor Coppola, thank you for the positive and constructive suggestions to improve our manuscript. Below are my responses to your comments..

*Comment 1: In the paper, the authors deal with the important issue of preferential flow, which has been widely analysed in the literature of the last 30-40 years. The work carried out in the paper consists of including the swelling-shrinkage process, induced by changes in soil water contents, in the well-known Dual Permeability Model (DPM) proposed by Gerke and van Genuchten (1996), to account for the dynamic changes of fractures volume with soil wetting and drying.*

*The approach used by the authors incorporates the swelling-shrinkage approach proposed by Stewart et al. (2016a, b) to describe change of porosity in both the soil fracture and matrix domains. The work is based on an important experimental work carried out in the laboratory on both small and relatively large columns filled with disturbed soil. The evolution of shrinkage with wetting and drying has been determined experimentally by analysing the images of the soil surface taken by a HD camera. Also, an improved exchange term proposed by Gerke et al. (2013) has been included to account for the exchange between the fracture and the matrix domains.*

*The data set coming from the column experiments has been used to evaluate the effectiveness of the simulations coming from the proposed model as compared to simulation coming from both a single domain and a rigid double domain model.*

**Response:** Thank you for this insightful summary of our work.

*General remarks*

*Comment 2: Based on my reading of the manuscript, the paper is quite well structured. I found everything quite clearly written and explained. The issue dealt with is clearly discussed in the Introduction of the manuscript, with an exhaustive literature. The figures summarize quite clearly the results. Some parameters in the tables should be described better. Also, a table with some more information on the correlation among fitted parameters should be given, along with their confidence*

*intervals. The materials and methods are well explained, with enough and clear details on the experimental work. The development of the fitting procedure is not completely clear. Results are complete and well-illustrated. Most of the discussion and conclusions coming from the numerical simulations seems well supported by the data.*

*Response: We are very happy with your appreciation of the work. We agree that some numerical parameters in Table 3 need further clarification. Please see reply to comment 10. Regarding to the fitting procedure, we will add more information in the revised manuscript.*

**Comment 3:** ① *To me, the issue dealt with in the paper is not novel.* ② *From a conceptual point of view, the paper mostly retraces the work already carried out by Coppola et al. (2012; 2015). Compared to the latter, the work under review incorporates a new approach for swelling-shrinkage changes of hydraulic properties in both the soil fractures and matrix, as proposed by Stewart et al. (2016a, b).* ③ *Also, the soil considered is a reconstituted soil, differently from the work by Coppola et al., who calibrated, validated and tested the model on data coming from experiments involving in situ undisturbed soil plot and undisturbed soils samples taken from the soil matrix of the same plot.*

**Response:** Thanks for these comments. For comment ①, yes, we agree with you that the PF-DC is a classical research topic, but still important to continuing studying as we believe the approach to simulate and quantify the PF-DC has room for improvement. In our paper, the novelty lies in the implementation of the Stewart et al (2016a, b) model for soil swelling-shrinking behavior, evolution of desiccation cracks and associated hydrological process during wetting-drying cycles.

For comment ②, it is true that we build on the pioneering work of Coppola et al (2012; 2015) as also referred to. In our work we show that the empirical relationships between the crack area and the suction head in our experiments do not follow a natural logarithm function. We realized that this could be ascribed to the different soil samples and different boundary conditions. In our work we follow the shrinking-swelling model proposed by Stewart et al. (2016a, b) which fitted our observation better. Hence, the Stewart et al (2016a, b) function was incorporated into the Richards-based dual-permeability framework, leading to a slightly modified dynamic PF-DC model for swelling-shrinking clays.

For comment ③, yes, you are correct, we used a reconstituted soil in our lab experiments. Using a reconstituted soil or an undisturbed soil has its own advantages and disadvantages. The former can eliminate the effects of other macropores on the preferential flow but needs long time to produce

highly developed desiccation cracks. The latter can provide well-developed desiccation cracks but may be affected by other macropores (i.e. root pores or earthworm holes).

**Comment 4:** *Some statements in the Introduction and in the discussion and conclusions seems a bit forced and misleading. I will try to argue about them, also to discuss some other issues the authors dealt with in the manuscript.*

*In the Introduction, the authors state: "Coppola et al. (2012); (2015) took another step forward to allowed crack volume and/or hydrological properties to vary as a function of soil shrinkage. However, the relationship proposed in the model, a natural logarithm function involving the suction head and crack proportion, lacks physical consistency with the variation of porosity. This implies a disconnection between hydrological properties and porosity in the crack domain."* ① **A similar statement may be found again in the conclusions of the paper under review. To me, this statement appears as a wrong and approximate interpretation and reproduction of a quite hurried conclusion drawn by Stewart et al. (2016a), who wrote (page 7912):** *"Coppola et al. [2012, 2015] allowed β and/or the soil hydraulic properties (e.g., volumetric water content, hydraulic conductivity) to vary as a function of soil shrinkage. However, the relationships proposed in those models lack physical consistency, in that domain specific hydraulic properties (e.g., hydraulic conductivity) remain constant regardless of changes in porosity distribution (e.g., β). This disconnect (as they wrote in their paper) between hydraulic properties and swelling ....".*

*As may be deduced in Stewart et al., the disconnection they speak about concerns the fact that hydraulic properties are not allowed to change with porosity changes. By reading carefully the paper by Coppola et al. (2012), this argument is unfounded.* ② **In the section 3.3, the authors clearly explained how the $\vartheta_a(h)$ and $K(\vartheta_a)$ ( $\vartheta_a$ is the moisture ratio of the soil matrix) experimental data points measured on the soil cores were converted to as many $\theta_a(h)$ and $K(\theta_a)$ points by using the equation 10a and the $e_a(h)$ a values measured at the same h. Thus, the $\theta_a(h)$ and $\theta_f(h)$ (and the corresponding $K(\theta_a)$ and $K(\theta_f)$) parameters comes from the $\vartheta_a(h)$ adjusted for the $e_a(h)$ data and, once used as input in the code, already account for the deformation of both domains with changing h. In other words, the θ and the K values calculated during simulations for a given h value at a given simulation node already accounts for the deformation of pore-size distributions of both the domains under swelling/shrinkage. What's more, the**

*authors also allowed the fraction of the matrix and fracture porosity to change along with hydraulic properties.*

*③ As this is not a simple task from an analytic point of view, they assumed a logarithmic function describing the β(h) evolution, but this is another story and has nothing to do with the physical inconsistency and the disconnection between changes in porosity and hydraulic properties Stewart et al. discussed about. As for the paper by Coppola et al. (2015), it simply showed three scenarios where the swelling-shrinking cycles were assumed to alternatively affect 1) only the hydraulic properties, 2) only the fraction of the two porosities with no effects on the hydraulic properties, 3) both, in the combined approach already presented in the 2012 paper. So, saying that these approaches do not account for changes in hydraulic properties is simply unfounded.*

**Response:** Thank you for pointing this out.

For comment ①, indeed we do have a statement looking similar as Stewart et al. (2016a), but our questioned point is not the same. Indeed, we disagree with their statement that "…in that domain specific hydraulic properties (e.g., hydraulic conductivity) remain constant regardless of changes in porosity distribution (e.g., β). This disconnect (as they wrote in their paper) between hydraulic properties and swelling …". We are aware that the Coppola et al 2012; 2015 model allows the hydraulic properties in each domain to vary with the porosity of each domain . We will explicitly mention this in our revision.

For comment ②, as shown in our text (Page 3 Line 93-95), we do not mention this issue. What we focus on is the physical consistency of the empirical relationships between the crack area and the suction head. Here we did not adopt the natural logarithm function as we argue that it may be not suitable to other soil types.

For comment ③, we also realized that our text created some ambiguity. We will change it as follows: Coppola et al. (2012); (2015) took another step forward to allow crack volume and/or hydrological properties to vary as a function of soil shrinkage. However, the relationship proposed in the model, an empirical natural logarithm function linking the suction head and crack proportion, could be not suitable to other kinds of soil.

*Comment 5: In any case, if the physical inconsistency lies in the disconnection between changes of porosity and corresponding changes in hydraulic properties, as argued by Stewart et al. (2016a, b),*

*this could more apply to the paper under review.* **① In fact, the authors should explain clearly where in their paper they change the hydraulic properties with swelling-shrinking cycles. If I well understood, their approach assumes fixed hydraulic property shape parameters (see table 3) for both the domains and the porosity is assumed to change according to equations 18 and 19. The Ks is scaled according to the changes in the porosity. I guess the saturated water content also scales similarly, even if I did not find any explanation of how the change in the porosity scales the water retention curves. Is Ks only a function of the porosity, rather than of the whole pore-size distribution? ② Do the authors believe that swelling-shrinkage simply scales the hydraulic properties (that is, swelling shrinkage has only effects on the total porosity of both the domains), as suggested by unchanged shape parameters of the soil hydraulic properties, or rather it changes the whole pore-size distribution (as considered by Coppola et al. in the 2012 paper and in the approach they called βk in the 2015 paper)?**

**③ To me, it seems that the argument of the physical inconsistency was introduced in the paper rather to maintain the usefulness of using a swelling-shrinkage approach a bit different from that previously used. I find the Stewart et al. approach actually effective and physically attractive, but this do not requires arguing that the approach by Coppola et al. is physically inconsistent.**

**Response:** Thanks for the insightful and thought-provoking comments.

For comment ①, indeed we fixed the SWRC shape parameters for each domain, and the matrix porosity is assumed to change with the saturation degree (from SWRC) by combining the shrinking-swelling parameters (Eq. 18 and Eq. 19). Our assumption means that the soil shrinking-swelling behavior has less influence on the SWRC shape but more influence on the porosity and therefore the saturated hydraulic conductivity. Hence, you're correct that in our model $K_s$ of each domain is only a function of the porosity.

For comment ②, we conceptualize the soil shrinking-swelling behavior, which has effects both on the total porosity and pore-size distribution. However, in this current study, we neglected the shift of pore-size distribution during shrinking-swelling process and assumed that process has more influence on the porosity variation. We will add more explanation involving our assumption in the revised manuscript.

For comment ③, we are sorry for the unwanted criticism and we will delete discussions related to the physically inconsistency.

*Other comment*

**Comment 6:** *Page 4, line 134: This means that the authors assume the $K_s$ and the hydraulic properties obtained on a reconstituted soil sample represent the $K_s$ and the hydraulic properties of the matrix in the reconstituted large soil column with the same bulk density. This is confirmed at page 14, line 397. This would imply that the $K_s$ depends only on the bulk density of the soil. The authors should discuss more this point;*

**Response:** Thanks for this fair comment. Here, please note that the $K_s$ obtained on a reconstituted soil sample represents the maximum $K_s$ of the matrix domain prior to shrinkage. When the soil matrix begins shrinking, the $K_s$ of the matrix domain will decline because its porosity decreases. The porosity of soil is negatively linear correlated to the soil bulk density. Therefore, you are correct that we conceptualize in our analysis that the $K_s$ depends only on the soil porosity. We will add this to the discussion.

**Comment 7:** *Table 1: I did not understand what is the optimal water content, $w_{opt}$, in the table;*

**Response:** Sorry for the short in explanation. The optimal water content, $w_{opt}$, is often used in the road engineering field, and it refers to the water content corresponding to the maximum dry density (also called the best compaction status). We will add more explanation in the revised manuscript.

**Comment 8:** *Page 7, line 221. This is the first time you introduce COMSOL in the paper. I would explain here what it is.*

**Response:** COMSOL Multiphysics is a finite element analysis solver and simulation software package for various physics and engineering applications, especially coupled phenomena and multiphysics. We will add this explanation in the revised manuscript.

**Comment 9:** *Page 12, line 337. Did the authors account for this interspace when calculating the fractures volume change?*

**Response:** We did not account for the interspace. As you can see in Fig. 3, to avoid pixel distortion near the photo edge, we only cropped central area of the photo to be used as crack image analysis.

**Comment 10:** *Table 3: as for the parameters pertaining to the DPM and DPMDy, are they the final parameters coming from fitting or are they the initial guess values? In any case, the fracture parameters seem quite strange. The saturated water content simply suggests a void without solid*

*particles, which is not coherent with the n parameter, suggesting at least a sand porous medium;*

**Response:** For Table 3, only SWRC parameters for the crack domain and mass transfer coefficient $a_w$ come from empirically guess. The other parameters all come from fitting. Regarding the SWRC parameters for the crack domain, we agree that the $n$ parameter looks a little bit small but we believe it may be still acceptable. Most importantly, it is the most robust value when running the model under wetting-drying cycles.

**Comment 11:** *Page 24, line 618. I did not understand the sentence "…improper exchange term (Coppola et al., 2012; 2015). Did these authors use an improper exchange term?*

**Response:** This indeed is misleading. As it can be seen in the manuscript (line 416-424), we mentioned that setting the $K_a$ as the arithmetic mean of hydraulic conductivity of the two domains would overestimate the $K_a$ when the hydraulic conductivity of the crack domain is much higher than that of the matrix domain. For instance, in our case, because we regarded the crack as empty space, its maximum $K_s$ is six orders of magnitude larger than that of the soil matrix. Therefore, the arithmetic form is improper under such conditions.

In Coppola et al. 2012; 2015, the crack domain was regarded as space filled with soil particles and the difference between the matrix and crack hydraulic conductivity fell within two orders of magnitude. Therefore, in that case, using the arithmetic mean was appropriate.

However, we hold that the crack domain should be regarded with a water storage space with large voids. Consequently, we state that the arithmetic mean leads to an improper estimate of the exchange term.

**Comment 12:** *Section 6.3. Model performance: It would have been interesting to see a table in this section summarising quantitatively the effectiveness of the model fitting, with optimized parameters, correlation matrix, confidence intervals, …*

**Response:** Thanks for the suggestion. We will try to add a table in this section.

---

## Author Comment (AC2)

**Reply to RC2's comments**

Dear Reviewer, thank you for the positive and constructive suggestions to improve our manuscript. Below are my responses to your comments.

*The present manuscript focuses on the experimental and modelling aspects of the hydrologic behavior of a shrinking soil under repeated wetting-drying cycles. In particular, the role of cracks and their dynamic behavior during the cycles are emphasized. The overall content of the paper is very interesting, relevant and fits within the aim and scope of the journal. Regarding the novelty aspects. Experimental: I am not an expert in the subject, but I am assuming (as it also transpires from the paper) that this type of experiments (e.g., setting, soil type, involved processes, scale of observation) are well-established; the experimental results are of quality. Modelling: the authors emphasize the novelty aspect of the proposed dynamic (i.e., varying with the saturation degree of the soil matrix) crack permeability, yet, going through the paper, it appears to me that the proposed model strongly leverages on previous formulations introducing then some assumptions. The reference to the literature seems appropriate and reach; Figures could be of a better quality; Writing: I am not a native English speaker, but there are several unclear parts in the text and several errors, I highly recommend a careful revision of the text.*
*I have a set of comments which I hope would help to improve the quality of the paper.*

Response: Thanks for your insightful summary of this study and we are very happy with your appreciation of our work. We agree that some of the figures and English text need further improvement. We will make revision in light of your suggestions in the new version. For the model novelty, please see reply to comment 1.

*Comment 1: the authors define the novel dynamic dual permeability model DPMDy mainly by setting the relative permeability of the crack always equals to 1 (Eq. (28) and line287) while they leverage on Eq. (27) proposed by Steward et al., 2016b to determine the absolute value of the permeability of the crack.* ① ***In this context, the DPMDy does not seem that novel, since it is mainly based on an assumption rather than a novel formulation/expression!*** ② ***Moreover, how reasonable it is to set the relative permeability of the crack always to 1? The less water there is in***

*the crack the smaller the crack permeability should be.*

*In section 6.2.1 the authors compare the values of crack permeability for the DPMDy and dual permeability (DPM) models (see Eq. (25)), see Figure 17: The striking feature being that the crack permeability for DPM decreases over the drying cycle, while that for the DPMDy increases.* ③ **_I would expect a decreasing trend for the crack permeability as the soil gets dry: we are speaking of the crack permeability associated with water, thus as less water is present in the crack the harder it gets to let it flow under a given head gradient; this is the meaning of including the relative permeability in Eq. (25)._** ④ **_At the same time, I do agree on the explanation provided by the author for the increasing trend of the DPMDy crack permeability: the drier the soil, the wider the crack, the easier it is to have water flowing … if we are talking of a completely saturated crack (as they assume), while I imagine that the saturation of the crack decreases during the drying cycle. I am seeing a bit of confusion on the meaning of absolute permeability, relative permeability and the permeability for a flowing phase._** ⑤ **_Moreover, I am wondering what would be the results (e.g., crack permeability to water) if Eq. (25) is combined with Eq. (27) (that provides the dynamic aspect of the absolute crack permeability to water)? DPMDy is Eq. (25) + Eq. (27) under the assumption of relative permeability of the crack to water fixed at one._**

Response: Thanks for these insightful comments!

For comment ①, we agree with you that fixing the relative permeability of the crack always to be 1 (abbreviated as "$K_{c,r} = 1$" hereafter) does not come from a novel formulation. Indeed, as one of the essential novelties in our model, we prefer to regard "$K_{c,r} = 1$" as a new strategy (or a trick). This strategy ensures the crack permeability varies with the crack aperture (or ultimately the matrix water content) instead of the crack saturation degree. Such a trick avoids the unreasonably decreasing trend of the crack permeability during the dying and enlarging process of desiccation cracks. This new strategy is useful and never been reported in other studies, thus we still hold it is an important novelty in our model.

For comment ②, you're correct that in microporous media (e.g., soil matrix), for which, in dry conditions, capillary potential is the dominant term to water potential energy, the less water, the smaller the soil conductivity. However, as we mentioned in our model (Line 262), the crack domain is mainly composed of large voids, through which water is assumed to flow according to Poiseuille law (laminar flow at atmospheric pressure). This assumption implies that crack conductivity

depends only on crack aperture, as the water content in that domain affects only the hydraulic radius of the flowing water cross section. The independence of crack conductivity from soil matrix water potential should not surprise, as it is a consequence of the non-equilibrium flow condition, which is typical of dual porosity media (e.g., Šimůnek et al., 2003), and which is responsible of the water exchange terms between the two pore domains (i.e., equation (3)). This exchange term would not exist if the potential energy of the water in the two pore domains was at equilibrium.

For comment ③, we are aware that in the usual unsaturated conductivity models, the relative permeability in Eq. (25) plays the role in linking the conductivity with the water content, and thus drying crack domain would always lead to decreasing crack conductivity. However, as mentioned above, such a trend would be physically unreasonable for the enlarging desiccation cracks, in which the assumption of Poiseuille laminar flow implies that the conductivity, in turn related to the hydraulic radius of the cross section of the water flowing through the crack, should grow with crack aperture, regardless the degree of saturation of the crack domain.

For comment ④, as already mentioned in the reply to comment ②, the assumption of laminar flow through the cracks, obeying Poiseuille's law, implies that the conductivity only slightly depends on the degree of saturation of the cracks, while it is directly related to crack aperture. This is the reason why, for the sake of simplicity, it is assumed to neglect the slight dependence of crack domain hydraulic conductivity on crack saturation degree, and consider only the dependence on crack aperture. Besides, the SWRC only controls crack domain water storage capacity, but it has no influence on the conductivity.

For comment ⑤, we have compared the curve of Eq. (25) + Eq. (27) to that of only Eq. (25). As shown in the figure below, the crack conductivity calculated by Eq. (25) + Eq. (27) would result much lower with respect to that only using Eq. (25), and it would also show a strange non-monotonic trend with the saturation degree.

[Figure]

Fig. reply-1 Crack permeability calculated by only using Eq. (25) and combing Eq. (25) + Eq. (27)

*Comment 2: Lines 596-597 "With regard to the water flux, while the three models all have acceptable errors to the measured data, the DPM overpredicted the water flux of PF-DC but underestimate the water exchange from cracks to soil matrix." It is my understanding that the DPM underestimates the water exchange from cracks to soil matrix w.r.t. to other models (e.g., see Figure 18 and Sec. 6.2.2), but not respect to the actual behavior which is not recorded in the experiment (it is a difficult task), please clarify.*

Response: Thanks for the comments! You're correct here. We will change this sentence in light of your suggestions as follow:

Line 596-597: With regard to the water flux, while the three models all have acceptable errors to the measured data, the DPM overpredicted the water flux of PF-DC but underestimated the water exchange from cracks to soil matrix compared to other models.

*Comment 3: What is depicted in Figure 16? The caption does not say it, a reader must search in the main text for it.*

*Use the same color legend for the two panels in Figure 8 (see measurements at 25 cm) and specify what are the additional data (red and blue curves), please. Figure 6: the legend is very small. General: I would avoid dashed (or dotted) curves when is not necessary (e.g., Fig. 6a; Fig. 12; Fig. 14; Fig. 15), the quality of the images is not very high and it gets quite hard to see dashed curves, please consider change them.*

Response: Thanks for pointing out these issues, we will revise these figures in our new version.

*Comment 4: Unclear text parts. Line 48 "the effects of crack dynamics on the PF-DC through experiment studies" should not be experimental? Line 51 "However, other studies found that the PF-DC also leads water to rapidly infiltrate into deep soil even desiccation cracks" even WHEN dessication? Line 59 "An improve understanding of the PF-DC combined with theory methods is also needed" THEORETICAL methods? Lines 66-67 "The DPoM and DPM concepts belong to the dual-domain framework that assumes the soil pore system can be represented" that assumes THAT the soil? Line 84 "volume and hydrological properties keep constant" remain constant. Lines 89-90 "Later modification of SWAP incorporated the aforementioned process, but with a cost of neglecting shrink-swell behavior of soil." A later modification .... but AT THE cost; Line 92: "Coppola et al. (2012); (2015) took another step forward to allowed crack volume…" to ALLOW. And many more throughout the whole text, e.g., Line 516 "In addition, another interesting phenomenon is the transient decrease of $\delta_{c,exp}$ and increase of 5cm $\delta_{exp}$ …" are you referring to $\delta_{exp}$ at 5 cm depth? It is not clear; Line 582 "It corresponds to some experimental results that the PF-DC also exists and leads water rapidly infiltrate into soils even desiccation cracks are nearly closed during … "   leads water TO rapidly infiltrate .... even IF dessication cracks are nearly. Please revise it very carefully!!*

Response: Thanks for pointing out the grammar issues. We will check through the manuscript to carefully revise these and other issues.

*Comment 5: SWRC at line 31 is not clear what it is. AOI in figure 3, what does it stand for? $S_{e,c}$ in Eq. (25) is not defined.*

Response: Thanks for pointing out. SWRC refers to the soil water retention curve and AOI indicates area of interest. $S_{e,c}$ is the saturated degree of the crack domain. We are sorry for presenting their abbreviation without any explanation at the first time when they appear in the manuscript. We will add detailed explanation of these abbreviations in the revised manuscript.

*Comment 6: After Eq. (1)-(4) the list of symbols is detailed by giving one line to each, this changes for Eq. (5)-(13), then again for Eq. (14) one line to each symbol. Be consistent!*

Response: Sorry for the mistakes. We will unify them as one line in the new version.

*Comment 7: Table 1 says statistical results, what statistics are involved here?*

Response: This indeed is misleading. We will revise "statistical results" as "manual readings".

*Comment 8: many parameters of the model(s) have been calibrated (see Table 3), but it is not clear how? Which calibration strategy has been used?*

Response: For Table 3, only SWRC parameters for the crack domain and mass transfer coefficient $a_w$ have been empirically assigned. All the other parameters come from fitting procedure to measured data. We will add more information about parameter estimation in the revised manuscript.

References:

Šimůnek, J., Jarvis, N. J., van Genuchten, M. T., and Gärdenäs, A.: Review and comparison of models for describing non-equilibrium and preferential flow and transport in the vadose zone, J. Hydrol., 272, 14-35, 10.1016/s0022-1694(02)00252-4, 2003.

---

## Author Response (AR3)

**Reply to Editor's comments**

*I do think the manuscript can benefit from an additional revision, which can be defined as minor, in my view. After receiving these, I do think I will be in a position to make my final assessment.*

**Response:**

Dear Editor,

Thanks for giving us the opportunity to make refinements to our manuscript. According to the reviewer's and your suggestions, we made the following revisions:

(1) Regarding the suitability of the V-G SWRC in the crack domain discussed by the reviewer, we added more explanations in the section "6.3 Model performance". See line 609-615 and below in direct response to prof Coppola. Basically, we agree that our conceptual approach to use VG for linking water potential with water content in a crack domain has limitations as one can argue capillarity has little effect in a cracked domain. But we argue there remains some capillarity. Moreover, we show with both our numerical analysis and supported with previous publications it is clear that with effective parameter settings the conceptual approach is quite suitable in practical conditions.

(2) We also made other two minor changes at line 261-265 and line 275 to better correspond our response to the suitability issue of the V-G SWRC in the crack domain.

Hope these revisions can meet with your approval.

**Reply to Professor Coppola's comments**

Dear Professor Coppola, thank you for the positive and constructive suggestions to improve our manuscript. Below are my responses to your comments.

*Comment: This is the second time I read the manuscript. To me, many of the issues I raised in the first review have been mostly solved. Some statements I found too strong have been smoothed.*

*The only problem still remaining in the paper is related to the fracture porosity, which in the approach of the authors, taken from Stewart et al. (2006) would completely consist of voids. This assumption allows finding an analytical relationship describing the changes of the macropore weight with shrinkage. Nevertheless, this is in contradicted in the text by the use of a van Genuchten-Mualem type hydraulic properties, with parameters that would indicate some gradual changes of water contents which may be only related to a porous system not simply consisting of voids. To avoid this contradiction, Coppola et al. (2012; 2016), assumed a true double porous system, thus with a macropore system being a true porous system, which can well be described by its own van Genuchten-Mulalem hydraulic properties. By doing that, the toll to pay was to assume a different shape of the β(h) relationship, depending on the fact that the shrinkage changes the fraction of the macropore system, the hydraulic properties, or both. The higher complexity introduced by these assumptions was to look for keeping some of the physical reality of changes induced by shrinkage.*

**Response:** Thank you for your insightful comments. Below are our response and also the revision to your comments:

Line 609-615: Secondly, the results support the suitability, in the crack domain, where capillarity has little effect, of V-G SWRC with effective parameters and a constant relative hydraulic conductivity ($K_r$=1). In fact, a common defect in classical DPMs is that they often set the hydraulic conductivity of the crack domain ($K_c$) varies as a function of the saturated degree calculated from the SWRC of the crack domain (i.e Eq. (25)). This will lead to an unreasonable extremely low $K_c$ in drying initial conditions (Aguilar‑López et al., 2020). Setting $K_r$=1 ensures that the magnitude of $K_c$ only depends on the crack area or the saturated degree of the soil matrix domain, which provides

a potential solution for remedying the shortcoming mentioned above.